



# How well are we able to close the water budget at the global scale?

Fanny Lehmann[1], Bramha Dutt Vishwakarma[1], and Jonathan Bamber[1]

[1]School of Geographical Sciences, University of Bristol

**Correspondence:** Fanny Lehmann (fanny.lehmann@bristol.ac.uk)

**Abstract.** The water budget equation describes the exchange of water between the land, ocean and atmosphere. Being able to adequately close the water budget gives confidence in our ability to model and/or observe the spatiotemporal variations in the water cycle and its components. Due to advances in observation techniques, satellite sensors, and modelling, a number of data products are available that represent the components of water budget both in space and time. Despite these advances, closure
of the water budget at global scale has been elusive.

In this study, we attempt to close the global water budget using precipitation, evapotranspiration, and runoff data at the catchment scale. The large number of recent state-of-the-art datasets provides a new evaluation of well-used datasets. These estimates are compared to terrestrial water storage (TWS) changes as measured by the GRACE satellite mission. We investigated 189 river basins covering more than 90 % of the continental land area. TWS changes derived from the water balance
equation were compared against GRACE data using two metrics: the Nash-Sutcliffe Efficiency (NSE) and cyclostationary NSE. These were used to assess the performance of more than 1600 combinations of the various datasets considered.

We found a positive NSE and cyclostationary NSE in 99% and 62% of the basins examined, respectively. This means that TWS changes reconstructed from the water balance equation were more accurate than the long-term (NSE) and monthly (cyclostationary NSE) mean of GRACE time series in the corresponding basins. By analyzing different combinations of the
datasets that make up the water balance, we identified data products that performed well in certain regions based on, for example, climatic zone. We identified that some of the good results were obtained due to cancellation of errors in poor estimates of water budget components. Therefore, we used coefficients of variation to determine the relative quality of a data product, which helped us to identify bad combinations giving us good results. In general, water budget components from the ERA5 Land and the Catchment Land Surface Model (CLSM) performed better than other products for most climatic zones. Conversely,
the latest version of the Catchment Land Surface Model, v2.2, performed poorly for evapotranspiration in snow-dominated catchments compared, for example, to its predecessor and other datasets available. Thus, the nature of the catchment dynamics and balance between components affects the optimum combination of datasets. For regional studies, the combination of datasets that provides the most realistic TWS for a basin will depend on its climatic conditions and factors that cannot be determined a-priori. We believe, the results of this study provide a roadmap for studying the water budget at catchment scale.





## 1 Introduction

A better understanding of hydrological processes at the catchment scale has been highlighted as one of the key challenges for hydrologists in the 21st century (Blöschl et al., 2019). One of the key processes is the terrestrial water cycle which can be described by the water balance equation,

$$\frac{dTWS}{dt} = P - ET - R. \tag{1}$$

This equation expresses the total amount of water gained by a river catchment in the form of precipitation (P) as a sum of, water returning back to the atmosphere through evapotranspiration (ET), water flowing out of the catchment in the form of runoff (R), and any changes in the terrestrial water storage (TWS). TWS is defined as the sum of water stored as snow, canopy, soil moisture, groundwater, and surface water (Scanlon et al., 2018). The water balance equation is a budget equation that follows the conservation of mass and it is an indispensable tool for validating our understanding of the catchment scale water cycle.

Several studies have used the water balance equation to explain hydro-climatic changes experienced in a river catchment (e.g., Landerer et al., 2010; Pan et al., 2012; Oliveira et al., 2014; Saemian et al., 2020), to validate modelled estimates of one component (e.g., Bhattarai et al., 2019; Long et al., 2015; Wan et al., 2015), or to estimate one component when others are known (Chen et al., 2020; Gao et al., 2010; Wang et al., 2014). It should be noted however that in these studies the accuracy of the result is limited by uncertainties associated to individual components. For example, Sahoo et al. (2011) attempted to
close the water balance equation for 10 large catchments and found that the imbalance error amounted to up to 25% of mean annual precipitation. Additionally, Zhang et al. (2018) highlighted the source of the imbalance error as being predominantly from stark disagreement between evapotranspiration estimates.

Obtaining high quality spatiotemporal estimates of components of the water balance is challenging due to a lack of global in situ measurement networks and political will to sustain any existing network. Therefore, the era of satellite remote sensing
offered an excellent solution to monitoring the hydrosphere. With the help of dedicated satellite missions, we are able to measure variables that can be used to estimate water balance components. However monitoring TWS has been the most difficult part since it includes water on and below the surface of the Earth, and optical remote sensing can only offer information near the surface. This issue was solved by the launch of a satellite gravimetry mission from GFZ and NASA in 2002, also known as Gravity Recovery And Climate Experiment (GRACE) (Wahr et al., 1998; Tapley, 2004). This mission measures the temporal
variations in the Earth's gravity field, which can then be related to water mass change on and below the surface of the Earth. GRACE provides the most accurate global estimations of TWS to date, which can be used in the water balance equation 1.

Another challenge concerns components like ET with a high spatial variability, which requires precise satellite estimates, not consistently available due to observational constraints (Fisher et al., 2017). Since ET accounts for up to 60% of precipitation in some regions, it is a crucial component of the water cycle (Oki and Kanae, 2006). It also constitutes the most significant
uncertainties of the terrestrial water cycle components (Rodell et al., 2015). The water balance equation has been used to compensate for this lack of knowledge and increase our understanding of ET. Water-budget studies have generally found that ET inferred from the water balance equation agrees well with remote sensing estimates in terms of seasonal cycle but presents





larger inter-annual variability (Liu et al., 2016; Pascolini-Campbell et al., 2020; Swann and Koven, 2017) and larger magnitudes (Bhattarai et al., 2019; Long et al., 2014a; Wan et al., 2015).

Apart from ET, our apprehension of R also benefits from water budget estimations. Although river discharge can be measured by gauges, the spatio-temporal coverage of in situ measurements is limited due to a lack of money in some regions and political will to share data. Uncertainties and biases in P have been found to be the main drivers of the inaccuracy in budget inferred R (Sheffield et al., 2009; Oliveira et al., 2014; Sneeuw et al., 2014; Wang et al., 2014; Xie et al., 2019). Water budget studies using R as a reference variable also point out the difficulty to find datasets able to close the water budget (Chen et al., 2020;

Gao et al., 2010; Lorenz et al., 2014). Moreover, ET and R are strongly intertwined and accurate estimates of one cannot be achieved without a better constraint on the other (Armanios and Fisher, 2014; Lv et al., 2017; Penatti et al., 2015).

To improve the reliability of available data, the water budget can be used as a discriminating tool to assess the accuracy of various datasets. For this to be achieved, there is a need to first evaluate the water budget closure globally, including basins of all sizes, and comparing as many state-of-the-art datasets as possible. This review is currently lacking because first, a majority

of studies have concentrated only on a few selected basins with specific climatic conditions (e.g. the Amazon basin, (Swann and Koven, 2017; Chen et al., 2020)) or highly impacted by human activities (e.g. the Yellow river basin, (Lv et al., 2017; Long et al., 2015)). Additionally, the studies which look at several basins worldwide have only evaluated sparsely distributed basins, which leaves entire zones without analysis (Sahoo et al., 2011; Pan et al., 2012; Lorenz et al., 2014; Liu et al., 2016; Zhang et al., 2018). This has deprived hydrologists of a comprehensive global overview of the water budget.

Returning to the requirement for basins of all sizes, basins were also generally chosen to be quite large in the majority of studies. It is known that the accuracy of GRACE measurements is directly proportional to the size of basin (Rodell and Famiglietti, 1999; Wahr et al., 2006; Vishwakarma et al., 2018), however the lower limit of $\sim 200,000\ km^2$ established by Longuevergne et al. (2010) and which has long been used is no longer a requirement to retrieve GRACE signals. It has been shown that basins as small as $\sim 70,000\ km^2$ can be precisely recovered by GRACE measurements and that their size do not

influence the closure of the water budget (Gao et al., 2010; Lorenz et al., 2014; Vishwakarma et al., 2018). They are therefore included in the current study.

Regarding the number of datasets to be examined, each water budget study uses different datasets, some of which were available only over a given continent or over short time periods. To the authors' best knowledge, Lorenz et al. (2014) conducted the study comparing the largest number of datasets by assessing more than 180 combinations of P, ET, and TWS datasets.

However, many datasets have since improved, especially reanalyses Era-Interim (Dee et al., 2011) and MERRA Land (Reichle et al., 2011). It would be beneficial to provide an updated evaluation of those widely used datasets.

The aim of the current study is thus to provide a revised overview of the water budget closure on a global scale. Section 2 presents the study area covering all parts of the globe (excluding Greenland and Antarctica) and the datasets. Then, section 3 details the metrics used to evaluate the water budget closure as well as the selection process for the best combinations.

Moreover, section 4 explains the results and discusses previous studies.



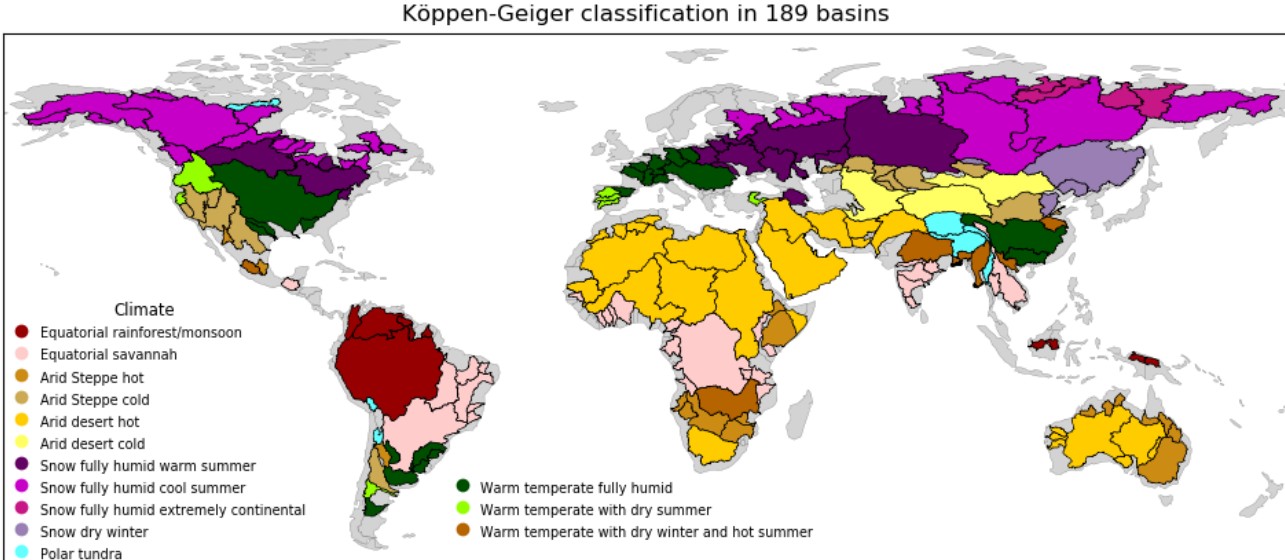

**Figure 1.** 189 basins larger than 63,000 $km^2$ with their corresponding climate zone

## 2 Data

### 2.1 Study area

We used the major river basins from the Global Runoff Data Centre (GRDC, 2020) to define the study area. Since the spatial resolution of GRACE products for hydrological applications is around 63,000 $km^2$ (Vishwakarma et al., 2018), catchments larger than this limit have been included in our analysis. Furthermore, these basins were assigned to a climate zone as defined by the Köppen-Geiger classification (Kottek et al., 2006). The 189 basins under study are depicted in Fig. 1 and their areas range from $\sim 65,600 \ km^2$ to $\sim 5,965,900 \ km^2$.

### 2.2 Datasets

We have used freely available global state-of-the-art datasets with a temporal resolution smaller than or equal to one month and coverage at least 2003 to 2014. If necessary, data have been interpolated to 0.5° x 0.5° grids using bilinear interpolation to correspond with monthly TWS derived from GRACE satellite mission. In this study, GRACE mascon fields were obtained from the Jet Propulsion Laboratory (JPL) RL06 (Watkins et al., 2015; Wiese et al., 2018). Our results were also computed with mascons from the Center for Space Research (CSR) and they can be easily reproduced with the code we provided. Since this did not significantly changed our findings, we only showed results using JPL mascons.

For other variables, daily data were aggregated to monthly values taking into account the number of days per month. Finally, gridded data were weighted by the area of each grid cell and then aggregated over a basin to obtain a time-series.



### 2.2.1 Precipitations datasets

Precipitation data were obtained from various sources that are summarised in Table 1. Three datasets rely only on rain-gauge measurements, namely the Climate Research Unit (CRU) which uses around 10,000 gauges (Harris et al., 2020), the Global

Unified Gauge-Based Analysis of Daily Precipitation from the Climate Prediction Center (CPC) based on approximately 30,000 gauges (Chen and Xie, 2008), and the Global Precipitation Climatology Centre (GPCC) maintaining a database of around 67,000 gauges (Schneider et al., 2020). Surface observations are often used to calibrate satellite estimations or as input variables in reanalyses. Since the global coverage of rain gauges is not homogeneous, the quality of such products varies regionally, thus satellite-based products provide a good alternative.

Two satellite missions were specifically designed to measure precipitation. The Tropical Rainfall Measuring Mission (TRMM) operated from 1998 to 2015 and provided monthly estimations of precipitation over 50° N; 50° S. We used the TMPA 3B43 version that extends TRMM measurements until 2020 via calibration with other satellites (Huffman et al., 2007, 2010). The Global Precipitation Measurement mission (GPM) was built on TRMM findings since its launch in February 2014. This constellation of satellites is calibrated using previous satellites through the Integrated Multi-satellitE Retrievals for GPM (IMERG)

to provide global coverage from 2000 onwards (Huffman et al., 2019). Finally, the Global Precipitation Climatology Project (GPCP) merges various satellite-based estimates with rain-gauge measurements from the GPCC (Adler et al., 2018). It provides a well-used and long dataset spanning from 1979 to the present.

Apart from these, reanalyses products provide consistent estimations of precipitation, evapotranspiration, and runoff. ERA5-Land is a rerun of the land component from the ERA5 reanalysis developed by the European Centre for Medium-Range Weather

Forecasts (ECMWF). Precipitation data are obtained from satellite measurements including but not restricted to TRMM and GPM results and are provided from 1981 onwards (Muñoz-Sabater, 2019). The Japanese 55-year Reanalysis (JRA55) also derives precipitation from satellite measurements with forecasts starting in 1958 (Kobayashi et al., 2015). Finally, the Modern-Era Retrospective Analysis for Research and Applications, version 2 (MERRA-2) uses two precipitation datasets from the CPC: the Global Unified Gauge-Based Analysis of Daily Precipitation described above and the Merged Analysis of Precipitation

which combines gauge-based and satellite measurements (Reichle et al., 2017).

Finally, two additional datasets that combine rain-gauge observations, satellite measurements, and reanalyses: the Princeton Global Forcing dataset (PGF) and the Multi-Source Weighted Ensemble Precipitation (MSWEP), were used in this study. PGF was included as it is one of the forcing variables used in the Global Land Data Assimilation System (GLDAS) (Sheffield et al., 2006). Recently developed, MSWEP merges gauge observations (including GPCC), satellite measurements (including

TRMM), and reanalyses (ERA-Interim and JRA55) (Beck et al., 2019).

Since there are large disagreements between different datasets, it is important to assess whether a dataset is in general agreement to others. By revealing datasets with significant bias, this method can limit the occurrence of error cancellation, which is a well-known problem in water budget studies (Sneeuw et al., 2014; Lorenz et al., 2014). We have used the coefficient of variation (CV) to evaluate various datasets of a water budget component in each basin. From a group of datasets, the CV is

a time-series defined as the standard deviation divided by the mean. (A minimum value of 10 mm was enforced for the mean



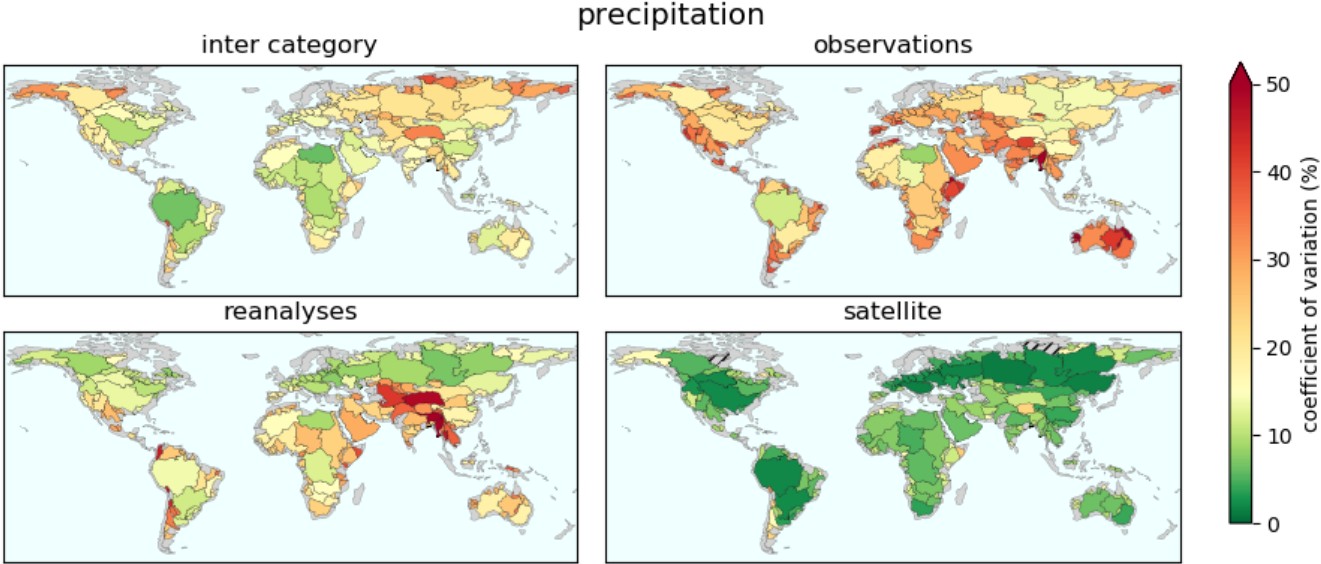

**Figure 2.** Coefficient of variation between different sets of precipitation datasets. Satellite: TRMM, GPM, and GPCP. Observations: CPC, CRU, and GPCC. Reanalyses: ERA5 Land, JRA55, and MERRA2. Inter-category: mean of satellite, observations, and reanalyses.

to avoid high CVs during the dry season). The higher the CV, the greater the disagreement between datasets. Figure 2 shows the mean of CV time-series in each basin. Unsurprisingly, satellite datasets (TRMM, GPM, and GPCP) provide close results since they use similar measurements and are therefore not at all independent. Observations datasets (CPC, CRU, and GPCP) are more independent, which leads to higher CVs. However, apart from Australia where CRU led to precipitations consistently

smaller than CPC and GPCC, there were no common patterns in the other regions. In addition, the major differences between reanalyses were found in Central Asia where MERRA2 gave much smaller precipitation values than ERA5 Land and JRA55. Interestingly, Fig. 2 also shows that the method used to create the dataset (i.e. rain gauge observations, satellite measurements, or reanalyses) is less discriminant than differences within a method. The inter-category CV measuring differences between the mean of observations, satellite, and reanalyses datasets was found to be relatively low. The highest CVs were found in

high latitude basins where reanalyses consistently led to larger precipitations whilst observations had the smallest precipitation values.

### 2.2.2 Evapotranspiration datasets

Evapotranspiration is the sum of evaporation from water surfaces and transpiration through vegetation. Datasets used in this study are listed in Table 2. One of the most accurate methods to estimate evapotranspiration is the Penman-Monteith equation

(Penman, 1948; Monteith, 1965). The variables used in this equation are obtained from various land surface parametrizations and energy balance equations in reanalyses ERA5 Land and MERRA2, and in GLDAS land surface models (LSMs). We chose





three variants of the GLDAS: the Variable Infiltration Capacity (VIC, Liang et al. (1994)), the Noah model (Chen et al., 1996; Koren et al., 1999; Ek et al., 2003), and the Catchment Land Surface Model (CLSM, Koster et al. (2000)). These LSMs are forced with different data depending on GLDAS version (Rodell et al., 2004). For example, PGF precipitation was used in

version 2.0, GPCP precipitation in version 2.1, and ERA5 precipitation in version 2.2 coupled with GRACE data assimilation (for CLSM only, (Li et al., 2019)). MOD16 algorithm also uses the Penman-Monteith equation with measurements from the Moderate-Resolution Imaging Spectroradiometer (MODIS, NASA) (Mu et al., 2011).

One of the main drawbacks of the Penman-Monteith equation is the reliance on a large number of parameters such as vegetation characteristics, air temperature, wind, vapour pressure, etc. Since these parameters can be difficult to assess accurately,

alternative approaches have been developed. For example, the Global Land Evaporation Amsterdam Model (GLEAM) uses an equation involving fewer parameters, the Priestley-Taylor equation (Martens et al., 2017; Miralles et al., 2011). Another method relies on the energy budget to compute the fraction of energy leading to water vaporization, as done in the Simplified Surface Energy Balance for operational applications (SSEBop), (Senay et al., 2013). Finally, algorithms also take advantage of the FLUXNET network of eddy-covariance towers measuring evapotranspiration. To this extent, the machine learning FLUX-

COM algorithm (Jung et al., 2019) extends the methodology of the well-used Multi-Tree Ensemble (Jung et al., 2009) by exploiting relationships between meteorological variables and latent heat flux measured by eddy-covariance towers.

Similar to precipitation, Fig. 3 shows the coefficient of variation for different categories of evapotranspiration datasets. CVs were relatively low between the mean of all categories, as was found for precipitation. The largest differences between reanalyses were also found in Central Asia with MERRA2 predicting lower evapotranspiration. In addition, it is striking to

see the large CVs among land surface models (CLSM, Noah, and VIC with versions 2.0 and 2.1). In this category, there were consistent patterns across all basins with VIC tending to underestimate ET while CLSM provided slightly larger values. The CVs were especially large in high-latitude basins due to low ET in the cold season. Moreover, in Fig. 3 we see that the differences between remote sensing datasets (FLUXCOM, GLEAM, MOD16, and SSEBop) are not spatially consistent. In Australia, MOD16 led to significantly lower ET, especially during the hot season (October to February). In South Africa,

differences were constant all year long with MOD16 being lower while FLUXCOM was rather high. We do not comment on CVs in hot deserts (Sahara, Arabian peninsula, and Central Asia) because FLUXCOM and MOD16 are not available in non-vegetated land areas.

### 2.2.3 Runoff datasets

Runoff is computed in LSMs as the excess water not evaporated from soils. This water infiltrates through the soil to the lowest

layers without communicating with adjacent grid cells. All LSMs presented above provide runoff estimates that were included in this study. River discharge measurements are also available from gauge records but they are not temporally consistent and heterogeneously distributed along rivers. To exploit these in situ measurements without the need to pre-process them, we used the recently developed machine learning GRUN dataset which provides runoff values at 0.5° x 0.5° spatial resolution from 1902 to 2014 (Ghiggi et al., 2019). This algorithm was trained with precipitation, temperature, and runoff measurements and

validated against independent river discharge observations from the GRDC.



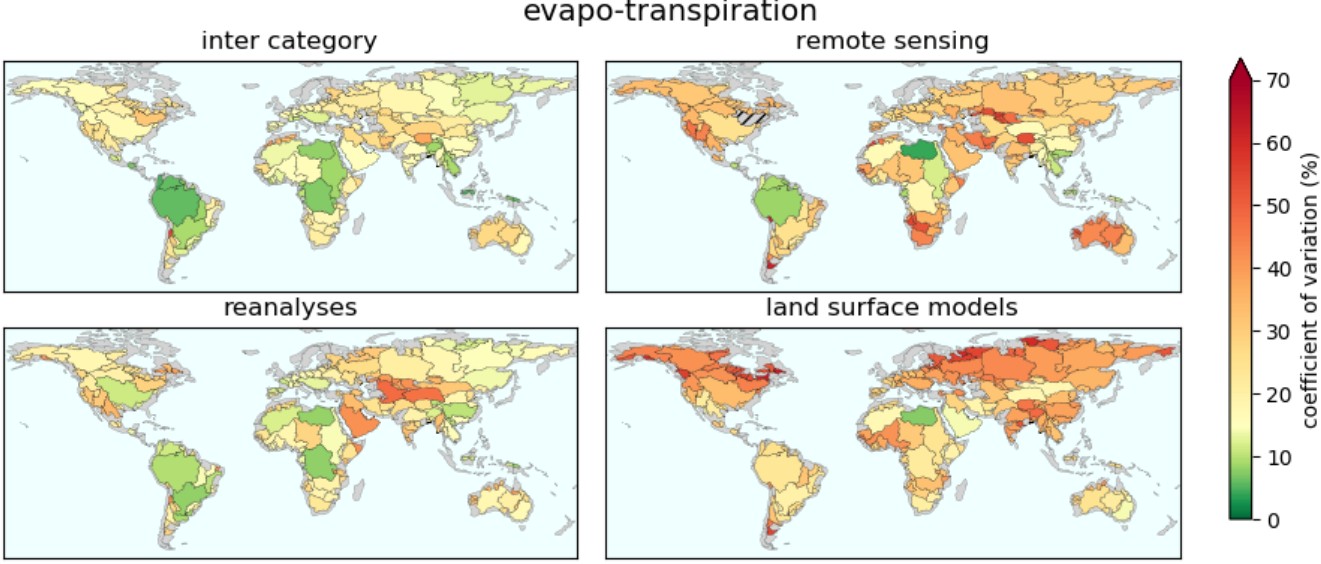

**Figure 3.** Coefficient of variation between different sets of evapotranspiration datasets. Remote sensing: FLUXCOM, GLEAM, MOD16, and SSEBop. Land surface models: CLSM, Noah, and VIC with versions 2.0 and 2.1. Reanalyses: ERA5 Land, JRA55, and MERRA2. Inter-category: mean of remote sensing, LSMs, and reanalyses.

As for precipitation and evapotranspiration, Fig. 4 shows the coefficients of variation. CVs were generally higher for runoff than that for evapotranspiration and precipitation. Even though it reflects high uncertainties in runoff values, this should play a relatively smaller role in the water balance because the runoff is the smallest water cycle component. In Fig. 4, the inter-category CVs were computed between GRUN, the mean of LSMs, and the mean of reanalyses. The general observations are complementary to those made about evapotranspiration. VIC generally led to the highest values among all datasets. Reanalyses tended to be lower, along with CLSM. Finally, compared with the mean across all datasets, GRUN was relatively close in general (not shown). The largest differences were found in Australia and Central Africa where GRUN was lower, and in Central Asia where it led to higher values.

## 3   Methods

### 3.1   Water budget reconstruction

GRACE mascon fields were used to compute time-series of TWS anomalies relative to the mean between 2004 and 2009. Since equation 1 involves the variation of TWS over a time period, which is called Terrestrial Water Storage Change (TWSC). To obtain TWSC from TWS anomalies, the time derivation was computed with centered finite difference (as in e.g., Long et al.





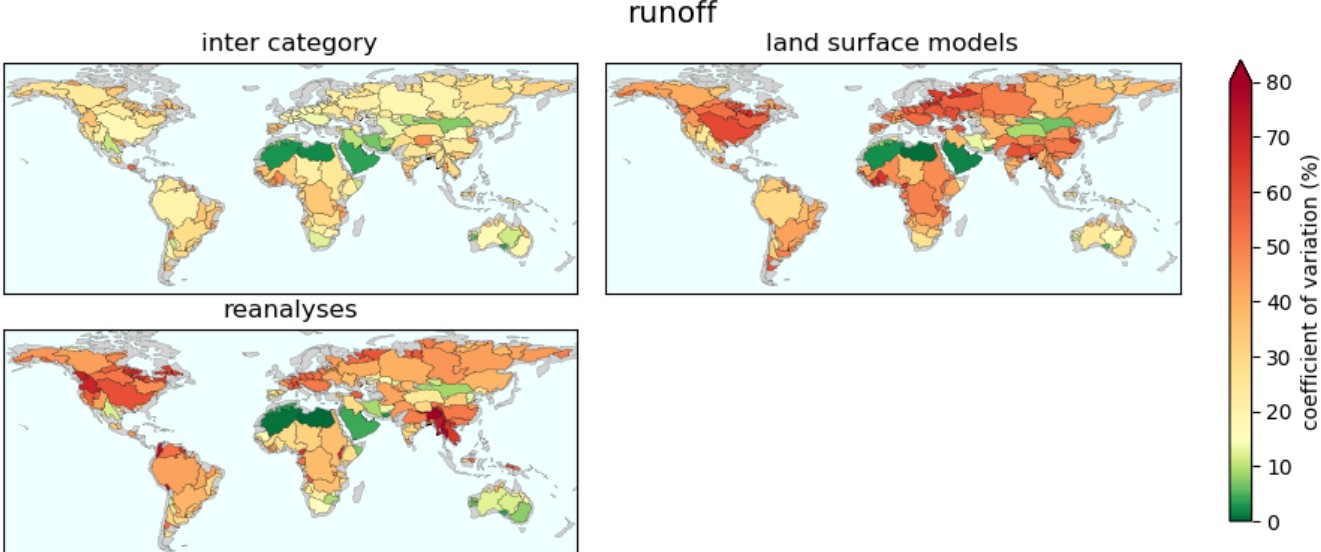

**Figure 4.** Coefficient of variation between different sets of runoff datasets. Land surface models: CLSM, Noah, and VIC with versions 2.0 and 2.1. Reanalyses: ERA5 Land, JRA55, and MERRA2. Inter-category: GRUN, mean of LSMs, and mean of reanalyses.

(2014b) or Pascolini-Campbell et al. (2020))

$$TWSC(t) = \frac{TWS(t+1) - TWS(t-1)}{2\Delta t}, \tag{2}$$

where $\Delta t$ equals 1 month and $t-1$, $t$, $t+1$ are three consecutive months. Missing monthly values were filled with cubic interpolation. In order to match the temporal shift induced by the central difference, time-series of P, ET, and R also needed to be time-filtered by equation 3 (Landerer et al., 2010)

$$\tilde{X}(t) = \frac{1}{4}X(t-1) + \frac{1}{2}X(t) + \frac{1}{4}X(t+1), \tag{3}$$

where $X$ denotes either P, ET, or R. All variables referred to hereafter are filtered variables but are denoted without the tilde notation for the sake of clarity.

Each triplet of datasets $(data_P, data_{ET}, data_R)$ was called a *combination* and led to a *budget reconstruction* of TWSC computed with equation 1: $TWSC_{budget}(t) = P_{data_P}(t) - ET_{data_{ET}}(t) - R_{data_R}(t)$. This reconstruction was compared with the derivatives obtained from equation 2 and denoted $TWSC_{GRACE}(t)$. Since we used 11 precipitation, 14 evapotranspiration, and 11 runoff datasets, we finally evaluated 1694 combinations.





## 3.2 Metrics

Differences between two time-series are commonly evaluated with the Root Mean Square Deviation (RMSD)

$$RMSD = \sqrt{\frac{1}{T}\sum_{t=1}^{T}(TWSC_{budget}(t) - TWSC_{GRACE}(t))^2}, \tag{4}$$

The main drawback of the RMSD is that it is not normalized *i.e.* basins with large TWSC tend to have larger RMSD. A very
common normalization is the Nash-Sutcliffe Efficiency (NSE) introduced by Nash and Sutcliffe (1970) to evaluate modeled
runoff compared to observations

$$NSE = 1 - \frac{\frac{1}{T}\sum_{t=1}^{T}(TWSC_{budget}(t) - TWSC_{GRACE}(t))^2}{\frac{1}{T}\sum_{t=1}^{T}(TWSC_{GRACE}(t) - \overline{TWSC_{GRACE}})^2} = 1 - \frac{RMSD^2}{err_{cst}^2}, \tag{5}$$

where $\overline{TWSC_{GRACE}} = \frac{1}{T}\sum_{t=1}^{T}TWSC_{GRACE}(t)$ is the long-term mean of TWSC and $err_{cst}$ is the deviation of monthly
values from the long-term mean. In our case, any positive value of the NSE means that the budget reconstruction of $TWSC_{GRACE}$
is a better approximation than the long-term mean. The maximum value of 1 describes a perfect reconstruction and a negative
value denotes a poor performance. One major advantage of the NSE is that it requires both phase agreement (usually assessed
with the correlation coefficient) and a small long-term mean error (evaluated with the bias, or percentage bias) to yield high
values (Lorenz et al., 2014).

However, although several attempts have been made to associate positive NSE values to a performance (e.g. Henriksen et al.,
2003; Samuelsen et al., 2015), it is known that this index suffers from several weaknesses, for example, a high positive NSE
can be obtained with a poor time-series if the time-series have a large variance (Jain and Sudheer, 2008). In the context of the
current study, basins with large seasonal variations of TWSC, especially tropical basins, are more likely to exhibit a NSE close
from 1 even though the budget reconstruction presents substantial errors.

To overcome this issue, it has been proposed to compare the budget reconstruction to the mean monthly value of TWSC
instead of comparing it to the constant long-term mean. The so-called cyclostationary NSE (Thor, 2013; Zhang, 2019) then
writes

$$NSE_c = 1 - \frac{\frac{1}{T}\sum_{t=1}^{T}(TWSC_{budget}(t) - TWSC_{GRACE}(t))^2}{\frac{1}{T}\sum_{t=1}^{T}(TWSC_{GRACE}(t) - TWSC_{GRACE}^m)^2} = 1 - \frac{RMSD^2}{err_{cyc}^2}, \tag{6}$$

where $TWSC_{GRACE}^m$ is the mean value for month $m$ over all years and $err_{cyc}$ is the deviation of GRACE TWSC from the
periodic monthly signal. Similarly to the NSE, positive values of the cyclostationary NSE indicate a budget reconstruction
better than the mean annual cycle, which measures the ability of the reconstruction to capture anomalous events (Lorenz et al.,
2015; Tourian et al., 2017).

Moreover, one can express the cyclostationary NSE in terms of the NSE by combining equations 5 and 6

$$NSE_c = \left(1 - \frac{err_{cst}^2}{err_{cyc}^2}\right) + \underbrace{\frac{err_{cst}^2}{err_{cyc}^2}}_{\gamma} NSE. \tag{7}$$



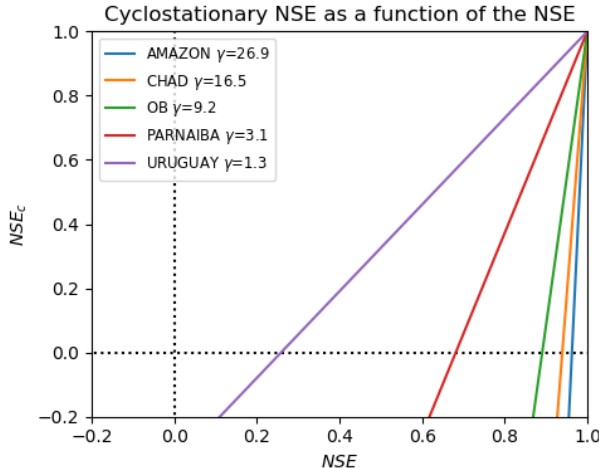

**Figure 5.** The cyclostationary NSE is related to the NSE through $NSE_c = 1 - \gamma + \gamma NSE$ where $\gamma = \frac{err_{cst}^2}{err_{cyc}^2}$

The $\gamma$ factor describes the behaviour of the TWSC by comparison with the mean seasonal cycle. Basins with periodic seasonal
cycles (i.e., low $err_{cyc}$) or large magnitudes (i.e., high $err_{cst}$) have larger $\gamma$. In those basins (e.g., the Amazon or Chad
basins in Fig. 2), extremely high NSE values are required to achieve a positive cyclostationary NSE, as can be seen in Fig. 5.
Special attention must then be given when examining such basins to discriminate performances depending on the NSE or the
cyclostationary NSE.

### 3.3  Selection of the most represented datasets

When estimating a water cycle component from the water balance equation 1, it is useful to know beforehand which datasets
are more reliable to close the water budget in the region under study. This section aims to describe how such datasets can be
selected. The NSE results were stored in a matrix where each row corresponded to a basin and each column to a combination.
Due to the matrix dimension ($189 \times 1694$), an automated computation was needed to evaluate the combinations. This was
achieved by introducing a cost function which represented the loss of accuracy when using any combination instead of the
optimal one.

Our method can be summarised as follows:

1. compute the cost matrix to describe the performance of each combination

2. cluster basins into larger zones depending on the similarities between cost vectors

3. for each zone, select the combinations satisfying a maximum cost and extract the underlying datasets

1. Using a cost function instead of the absolute metrics allowed us to overcome the lack of a NSE scale. On the one hand,
there are significant differences between a combination leading to a budget reconstruction with a NSE close to 0 and another



leading to an almost perfect reconstruction (NSE close to 1). These differences can be seen for example in terms of months where the budget reconstruction is within GRACE confidence interval. Therefore, we want to favour combinations leading to the highest NSE values. On the other hand, one cannot determine a NSE threshold assuring a satisfying reconstruction in

all basins. Figure 5 shows that very high NSE values were needed in basins with large $\gamma$ to outperform the monthly periodic signal. Consequently, a cost function evaluates the performances of a combination relatively to the largest NSE achievable in each basin. The cost function was then defined from the NSE by

$$c_i^b = \max_{comb} NSE^b(comb) - NSE^b(combination_i), \qquad (8)$$

where the maximum was computed over all 1694 combinations. We emphasize that the cost was evaluated independently for

each basin (denoted by the superscript $b$), allowing the maximum NSE to be different in each basin. For combinations leading to a cost larger than 2 (*i.e.* a NSE below -1), the cost was restricted to 2. This limited the penalization of combinations with highly negative values but had no major influence on our results since we focused on the best performing combinations.

2. From the cost matrix, each basin could be represented by a vector of 1694 costs. The similarities between two basins $b_1$ and $b_2$ were evaluated based on the Euclidean distance between their respective cost vector, $d(b_1, b_2) = \sqrt{\sum_{i=1}^{1694}(c_i^{b_1} - c_i^{b_2})^2}$.

For two basins to have a small Euclidean distance, each combination $i$ should lead to a similar cost in all basins: either the combination was satisfying in both cases ($c_i^{b_1} \simeq 0$ and $c_i^{b_2} \simeq 0$), or it did not perform well in both ($c_i^{b_1} \simeq 2$ and $c_i^{b_2} \simeq 2$). A hierarchical clustering algorithm was then applied to cluster basins so as to minimize the variance between cost vectors inside a cluster (Mueller et al., 2011).

3. Finally, the maximal cost for combinations to be considered as satisfying the water budget closure was chosen to be 0.1.

This means that the difference between the RMSD of a suitable combination and the lowest RMSD over all combinations is in average lower than $A/10$ where $A$ is the mean seasonal amplitude of TWSC. This threshold guarantees that selected combinations have performances similar to the optimal combination. Then, in each cluster determined by the algorithm, we selected the combinations with a cost lower than 0.1 for all basins in the cluster. From the selected combinations, we extracted the underlying datasets of P, ET, and R. By reporting the number of combinations in which each dataset appeared, we could

evaluate whether a dataset was clearly better than the others in a given region.

## 4 Results and discussion

### 4.1 Water budget closure

In order to assess the global water budget closure, we first examined the best performances across all combinations. This means that for each basin, we reported the highest NSE among all 1694 combinations. Figure 6 shows the maximum NSE that can be

achieved from a combination. Please note that a positive NSE was obtained over 99% of the total study area. Only 9 basins out of 189 did not achieve a positive NSE for any combination. They were mainly hot arid deserts in Northern Sahara, Somalia, Australia, as well as two other basins in Papua New Guinea (Mamberamo basin) and Hayes basin (Canada) (Fig. 6). The poor performances in arid basins can be explained by limited precipitation and water storage variations that lead to a low signal-to-

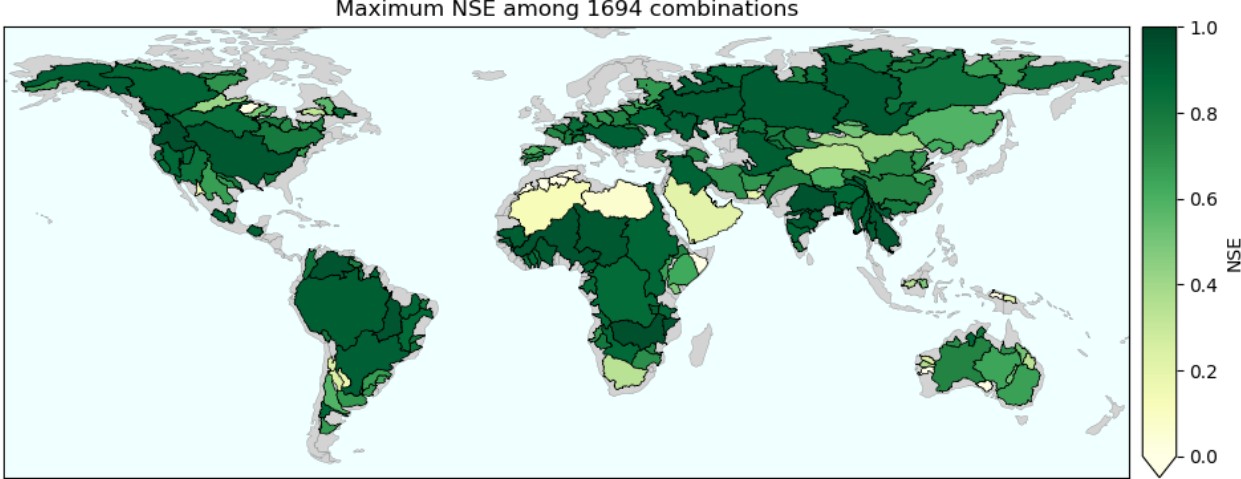

**Figure 6.** Maximum NSE per basin over all combinations. Green positive values mean that the budget reconstruction is a better approximation of GRACE TWSC than the long-term mean.

noise ratio. This is a major difference from previous studies where for example, Lorenz et al. (2014) found that only 29 basins
out of 96 achieved a positive NSE.

Figure 6 can be interpreted as follows: all the basins with a positive NSE offer a budget reconstruction better than the long-term mean from GRACE TWSC. In addition, higher NSE values correspond to a better fit between reconstructed TWSC and GRACE TWSC. Figure A1 then shows the distribution of the maximum NSE. Although it has been explained in Section 3.2 that positive NSE should be interpreted cautiously, one can observe that 61% of the study area satisfied a NSE larger than 0.8
which is usually considered as very good performance (*e.g.* Henriksen et al. (2003), Samuelsen et al. (2015)). Being given the large number of datasets, it is likely that cancellation of errors explains some of the good performances. The reader should remain cautious about this possibility when trying to reproduce our results and may use discrepancy measures such as the CV to examine datasets, as is explained in the following sections.

From its definition, the NSE can only be used to compare the budget reconstruction with the long-term mean. Since pre-
dicting intra-annual variations of TWSC would be more beneficial for hydro-meteorological studies, the cyclostationary NSE was also used to assess the quality of reconstructed TWSC. Figure 7 shows that a positive maximum cyclostationary NSE was achieved over 62% of the study area. It means that in those basins, the reconstructed TWSC was better than the mean annual cycle obtained from GRACE TWSC. The budget reconstruction performed especially well in the continental United States and Central America, in most of Southern America except the Amazon and the Andes, in Southern Africa, Australia, Europe, West
Russia, and East Asia (Fig. 7).

When comparing Fig. 6 and Fig. 7, one can observe that despite a very high NSE, some basins could not reach a positive cyclostationary NSE. This happened especially in tropical basins like the Amazon, some catchments in Western Africa, India,



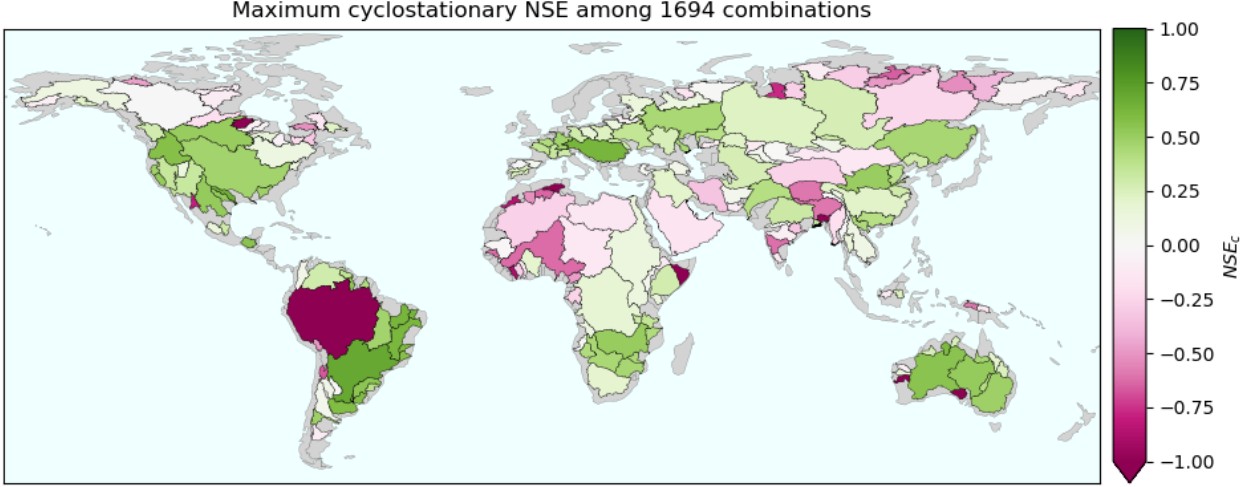

**Figure 7.** Maximum cyclostationary NSE per basin over all combinations. Green positive values mean that the budget reconstruction is a better approximation of GRACE TWSC than the mean monthly values.

and Myanmar. These basins illustrate i) the limits of the NSE and ii) the need for a complementary metric to evaluate the reconstruction. These two points corroborate the conclusions of Jain and Sudheer (2008). The Amazon basin exemplifies why

the NSE should not be used alone to assess the water budget closure. In fact, even with the best combination, the budget reconstruction consistently underestimated the magnitude of TWSC (Fig. A2). TWSC was too low in the wet season (January-March) and too high in the dry season (July-August). This indicates that the budget reconstruction was not good enough to capture the inter-annual as well as annual variability in TWS. Due to the large amplitude of TWSC in the Amazon basin ($[-100; 100$ mm/month$]$), the NSE was still very high ($maxNSE = 0.91$) and could mislead us into concluding that the budget

reconstruction is excellent. However, when assessing the cyclostationary NSE ($maxNSE_c = -1.28$), it appeared that the mean monthly values were a better fit to GRACE values than the budget reconstruction (Fig. A2).

The underestimation of annual variability in TWSC can be seen in the correlation plot between GRACE TWSC and our approximation (Fig. A3). Due to the error in approximating the largest TWSC, the regression slope is 0.7, while 1 is the optimal value. Figure A2 additionally shows that the water balance error is larger than GRACE uncertainty in 21% of months,

meaning that the error is significant.

However, one should not conclude that all basins with a high NSE and negative cyclostationary NSE exhibit the same behaviour. The Niger basin is indeed another basin with a high NSE (0.94) and a negative cyclostationary NSE (-0.62). Contrary to the Amazon, there was no consistent pattern in the water closure error and the error was lower than GRACE uncertainty in 94% of months (Fig. A4). The regression slope was also almost perfect as shown in Fig. A5. In such a basin with low

inter-annual variability, the error between GRACE TWSC and the mean monthly signal is very low (RMSD=6.6 mm/month). Therefore, achieving a budget reconstruction more accurate than the monthly signal may be an unrealistic expectation.





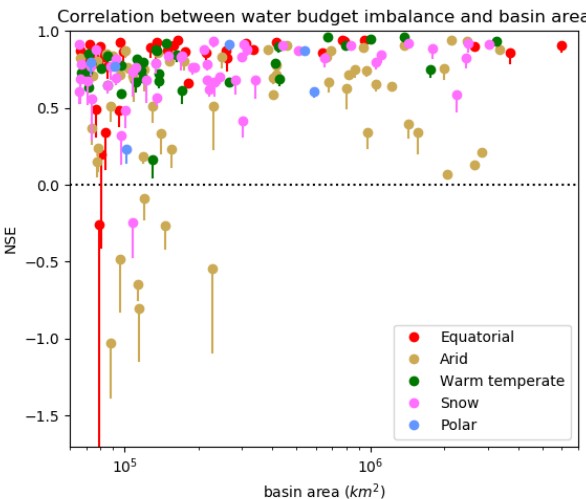

**Figure 8.** Each basin is represented by a bar between the maximum NSE (dot) and the 10th highest NSE

In conclusion, while the cyclostationary NSE is useful to assess intra-annual variations in the budget reconstruction, it is not the best assessment tool for all the tropical basins with almost periodic TWSC. The regression slope between the reference and approximate TWSC can help in exhibiting consistent patterns in the water balance error.

**4.2    Variables influencing the water budget closure**

Several studies have limited their budget computation to large catchments only due to the general notion that the accuracy of budget closure increases with the size of the basin. We found that both small and large basins can achieve a high NSE (*cf.* Fig. 6). Furthermore, Fig. 8 proves that there is indeed no correlation between the maximum NSE and the basin area ($R^2 = 0.12$, $p = 0.12$). Although limiting their study to 10 large river basins worldwide, Sahoo et al. (2011) found no relationship between
budget closure error and basin size. We extend this result and show that basins as small as 65,000 $km^2$ can close the water budget. This result still holds if we evaluate the correlation between the basin area and the maximum cyclostationary NSE ($R^2 = 0.01$, $p = 0.90$).

Figure 8 additionally indicates the consistency of our findings. Each basin was represented by a bar between the highest and 10th highest NSE values and the length of the bar was lower than 0.15 in 90% of the basins. This means that several
combinations were able to close the water budget with similar imbalance errors.

Additionally, basins can be classified depending on their climate zone. Figure 9 shows the distribution of the maximum NSE in each climate zone. Since the boxes (interquartile range) are of limited length (except for 'equatorial rain forest/monsoon' and 'hot arid deserts'), this suggests that the imbalance error is rather consistent inside a given climate zone. In 'equatorial rain forest/monsoon' climate zone, basins generally reached higher NSE values (map 6). However, this zone also contains small
Pacific islands (Papua New Guinea and Borneo) where runoff is much more important than evapotranspiration. Tables A1 and

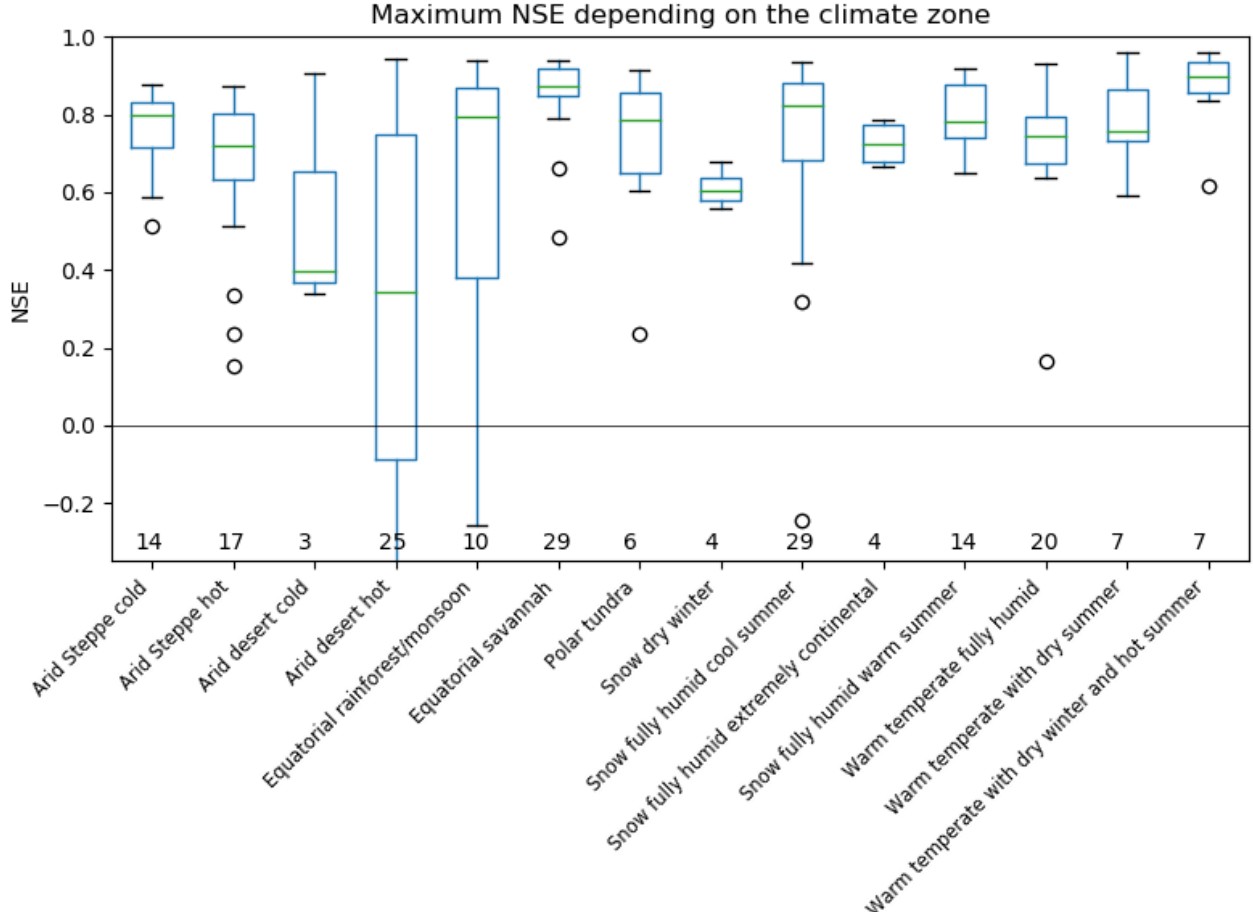

**Figure 9.** Boxplot of the maximum NSE per climate zone. The green line indicates the median, the box extends from the 1st quartile ($Q_1$) to the 3rd quartile ($Q_3$) while whiskers go from $Q_1 - 1.5(Q_3 - Q_1)$ (or the minimum value if higher) to $Q_3 + 1.5(Q_3 - Q_1)$ (or the maximum value if lower). Circles denote basins lying out of the whiskers. Figures represent the number of basins in each climate zone.

A2 indicate that runoff was more uncertain (around 30% disagreements between datasets) than evapotranspiration (around 18%) in those basins. Pacific islands with large runoff thus probably suffered from the poor runoff quality which led to low NSE values.

Hot arid deserts also have a large spread in the water budget imbalance (Fig. 9). Among those basins, some were entirely desert (Arabian peninsula, Sahara, Somalia, South, and West Australia) with a low signal-to-noise ratio, as previously mentioned. Other basins were partially covered by steppe (Australia, Orange, around Indus) or equatorial savannah (Niger, Chad, Nile). In those basins, precipitations occurred in the more humid subregions, thus increasing TWS variations. As a consequence, the error in the datasets became less significant and allowed a proper budget reconstruction.





### 4.3 Overall combinations performances

Although a majority of basins achieved a positive cyclostationary NSE, they differed greatly in terms of the number of combinations yielding positive values. As an example, 839 combinations satisfied a positive $NSE_c$ in the Sao Francisco basin while only 94 did so in the neighbouring Tocantins basin (Fig. A6). Therefore, we wanted to evaluate the ability of a single combination to close the water budget worldwide. To do so, we evaluated the total area of basins with a positive cyclostationary NSE for each combination. Table 3 shows the 20 combinations leading to the largest area.

It appears that choosing all three variables (P, ET, and R) from ERA5 Land yields significantly better results than the other combinations (35.5 million $km^2$ with a positive $NSE_c$ from the total study area of 96.6 million $km^2$). Figure 10 indicates that ERA5 Land performed well in the Central and Eastern United States of America (USA), but it failed to provide the positive $NSE_c$ of Fig. 7 in the mountainous Western basins (Columbia, Great basin). Again comparing with the best possible results, ERA5 Land performed quite poorly in the equatorial region of South America (Amazon basin and above), in Central Eurasia
(around the Ob, Aral sea, and Indus basins), as well as in several basins in Europe.

Knowing that there exists at least one combination giving a positive cyclostationary NSE in 62.3 million $km^2$, Table 3 shows that even the best combinations were far from approaching this number. This confirms that it is for now clearly impossible to achieve a good water budget closure with a single combination (Gao et al., 2010; Lorenz et al., 2014).

The second best combination in terms of area satisfying a positive cyclostationary NSE was the Catchment Land Surface
Model (CLSM) forced with version 2.0 of GLDAS (in particular PGF precipitations). Table 3 shows that 30.8 million $km^2$ reached a positive $NSE_c$ with this combination. Similar observations as ERA5 Land can be made generally, with good performances in Central and Eastern USA, South East America, and Australia. CLSM2.0 was more consistent than ERA5 Land in Europe but less so in Africa.

When looking at the following combinations, it appeared that their performances were more similar, compared to the dif-
ferences observed between the two best combinations. Table 3 also shows that each variable has a determining impact on the water budget closure. Indeed, choosing for example CLSM2.2 for runoff instead of ERA5 Land (as shown in the left column of Fig. 10) led to poorer results in Alaska, Asia, and central Africa while it improved NSE values around the Amazon basin.

Concerning GLDAS LSMs, it is clear in Table 3 that CLSM was a globally better LSM than Noah and VIC. We also noted that when using all variables from the same LSM, GLDAS 2.0 was globally better than version 2.1 for all LSMs (CLSM,
Noah, and VIC). As illustrated in the right column of Fig. 10, major differences are observed in Europe, Western Russia, and Alaska. This can be explained by disagreement between precipitations from GPCP and PGF. For instance, CLSM2.1 yielded only low NSE values in most of Eastern Europe whereas version 2.0 of the same model achieved a positive cyclostationary NSE. This last finding reflects the conclusion of e.g., Mueller et al. (2011) and Zaitchik et al. (2010) that forcing variables have a considerable influence on land surface models outputs.

We also point out that the ranking in Table 3 was not significantly modified by discriminating basins on the area satisfying a NSE larger than 0.5 (usually considered as good performances) instead of a positive cyclostationary NSE. This ensures the reliability of the method used to highlight the most consistent combinations.





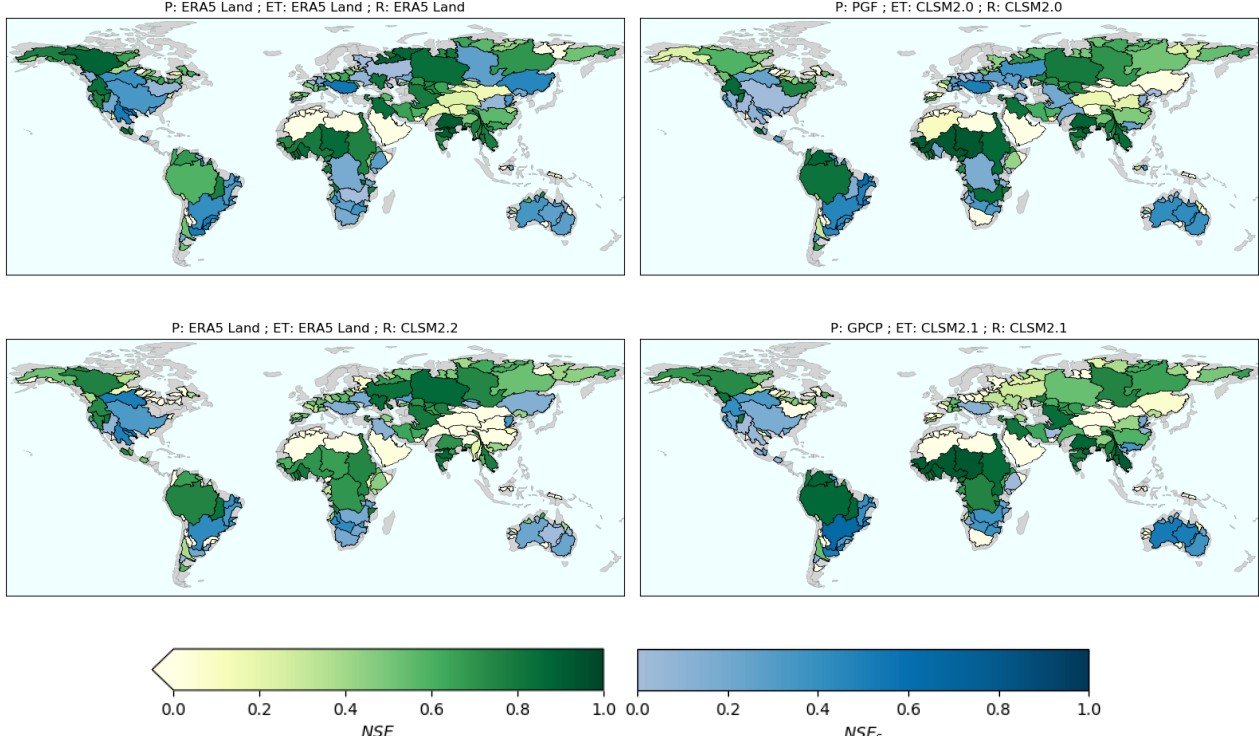

**Figure 10.** NSE and cyclostationary NSE with the first combinations in Table 3. Basins with a positive cyclostationary NSE are represented with blue shades corresponding to the $NSE_c$. Remaining basins are depicted in green, according to their $NSE$.

## 4.4 Datasets suitable in given regions

In the previous section, numerous combinations of global datasets were evaluated. This section aims to describe regions where
some datasets are more suitable than others to close the water budget. In a given basin, we defined as *suitable datasets* those appearing in combinations leading to a cost (difference between the maximum NSE and the NSE for a specific combination) lower than 0.1. This threshold was chosen to ensure that only the highest performing combinations were considered as suitable. We study only 132 basins out of 189, those with an excellent budget closure (maximum NSE larger than 0.8 or maximum $NSE_c$ larger than 0.1).

In general, many combinations were below the maximum cost: at least 112 combinations were suitable in 50% of the basins, at least 185 combinations in 25% of the basins. For a detailed review of suitable datasets in each basin, the reader is referred to Fig. A12, A13, A14, and A15. Although there was a large choice of combinations to close the water budget, two basins with similar characteristics only had a few suitable combinations in common. This makes a global and comprehensive evaluation of datasets more complex.





In addition, we observed that suitable datasets in a basin could generally not be mixed, suggesting that some cancellation bias occurred. As an example, Fig. 11 shows that suitable datasets in the Mississippi basin have considerably different seasonal cycles. Combining a precipitation dataset with high amplitude (GPCP) with low runoff (CLSM2.2) could close the water budget if associated with a high evapotranspiration (CLSM2.1, leading to $NSE_c = 0.32$) but not with a low evapotranspiration (Noah2.0, $NSE_c = -1.8$). Since there is no reason to consider a dataset as more reliable than others in the absence of unbiased

observations, care must be taken when combining suitable datasets.

        In order to provide a general overview of datasets performances, we choose to gather basins achieving the water budget closure for similar combinations. Those regions were determined with the hierarchical clustering described in section 3.3. The 132 selected basins with a good water budget closure are depicted in the dendrogram Fig. A7 and clusters represent basins with similar costs for the same combinations. We chose 13 such clusters comprising major basins of the world to provide a precise

but as succinct as possible overview of the datasets' performances. These clusters are denoted by the colored lines in Fig. A7 and are shown with the same basin colors on the map in Fig. 12.

        Basins clustered together in the dendrogram Fig. A7 were either neighbouring basins (e.g., Eastern Europe or Eastern Australia) or basins with similar geographical conditions. It is therefore sensible that the same combinations performed well in those basins. Among basins with similar characteristics, we pointed out large rivers in temperate regions (Mississippi, Parana,

and Danube basins) or cold basins with different snow conditions (Yenisei, Lena, Mackenzie, Yukon, and Kolyma basins).

        For each of the 13 clusters, we selected combinations yielding to a cost lower than 0.1 in every basin of the region. Figure 12 shows which datasets can be used in combination to satisfy the water balance. It first appears that among the precipitations datasets, the rain-gauge-based GPCC was often found in combinations satisfying the maximum cost, along with the satellite-augmented GPCP, reanalysis ERA5 Land, and the multi-source PGF. As a first approximation, those datasets are suitable for

global water budget analyses. However, for regional analyses, a closer look at individual datasets is required to obtain all possibilities.

        Figure 13 (top left) shows the decay in NSE when using GPCC as the precipitation dataset. It confirms that GPCC was very close to the best-performing precipitations datasets. Surprisingly, Fig. 13 also indicates that although GPCP added satellite measurements to GPCC observations, it increased the water budget imbalance in Eastern Europe and western Russia, as well

as in Congo and South Africa. GPCP performed notably well in South America, along with ERA5 Land that was one of the most consistent datasets for precipitation. The only region where ERA5 Land was not suitable was around China and Saint Lawrence basin. As shown in Fig. 12, PGF precipitation were able to close the water budget predominantly in Europe, as well as in Central Africa.

        For comparison, Fig. A8 indicates that CRU which never appears in the map 12 performed very poorly compared to other

datasets. Harris et al. (2020) mentioned that no homogenization of data was performed in CRU data. It also uses climatology values when measurements are missing, making it more appropriate for global analyses. The other rain-gauge-based dataset CPC was mainly suitable in Europe and China (see Fig. 12). Since MERRA2 is based on CPC observations (except in Africa where slight variations can be seen in Fig. A8), similar conclusions can be drawn for MERRA2. In addition, using GPM instead of TRMM (where we recall that GPM includes and extends TRMM results) improved the water budget closure. Finally, there





**Figure 11.** Datasets appearing in suitable combinations in the Mississippi basin (cost lower than 0.1). The discrepancy is similar to the coefficient of variation, except that the numerator is the difference between the maximum and minimum values instead of the standard deviation.

was no overwhelming advantage in choosing the multi-source MSWEP dataset. It is consistent in Europe and South America but should be avoided in snow-dominated regions of Eastern Russia and Alaska (Fig. A8).

Fig. 12 clearly shows that evapotranspiration from the land surface model VIC should be chosen in Russian snow-dominated basins, with a preference for version 2.0 compared to 2.1. However, this dataset should not be used in hotter regions such as South America, Africa, or Australia (Fig. 14). We found that VIC produces smaller evapotranspiration than other datasets,



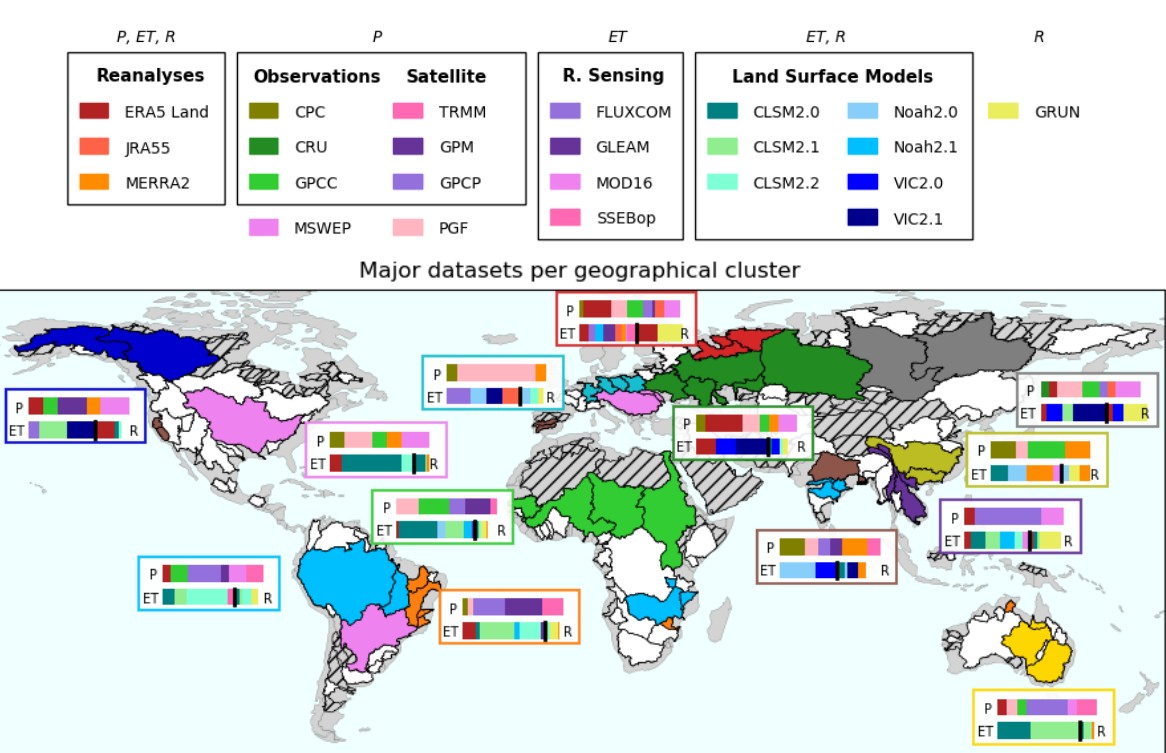

**Figure 12.** Datasets appearing in combinations that satisfy a cost lower than 0.1 for all basins inside the cluster. The 13 clusters highlighted in Fig. A7 are shown with different colors. For each cluster, the top line of each box represents precipitation datasets. The left part of the bottom line is evapotranspiration datasets while the right part is runoff. The limit between ET and R is symbolized by a black line located proportionally to the portion of ET in the mean annual water cycle of the corresponding region. Hatches show basins with a poor water budget closure (maximum NSE lower than 0.8 and maximum $NSE_c$ lower than 0.1).

along with higher runoff. The Catchment Land Surface Model was also consistently found in Fig. 12. Version 2.0 and 2.1 performed similarly (except in Europe where version 2.0 was better as already mentioned) and were especially suitable in equatorial (South America, Subsaharan Africa, Australia) and some temperate regions (South-Eastern Europe and the USA). Similar to precipitations, ERA5 Land evapotranspiration is an excellent dataset in most of the regions except the Amazon basin, China, and Australia (Fig. 14).

Evapotranspiration from version 2.2 of the Catchment Land Surface Model provided a good water budget closure in most of South America, Europe, and especially South Asia. However, it led to unrealistic low values in snow-dominated basins (see Fig. 14). An example of this behaviour is given in Fig. A10 where highly negative values appear in autumn. Since this dataset assimilates GRACE measurements and was validated against GRDC observations, this may reflect overfitting of runoff that is better constrained than evapotranspiration, therefore leading to unrealistic ET values.

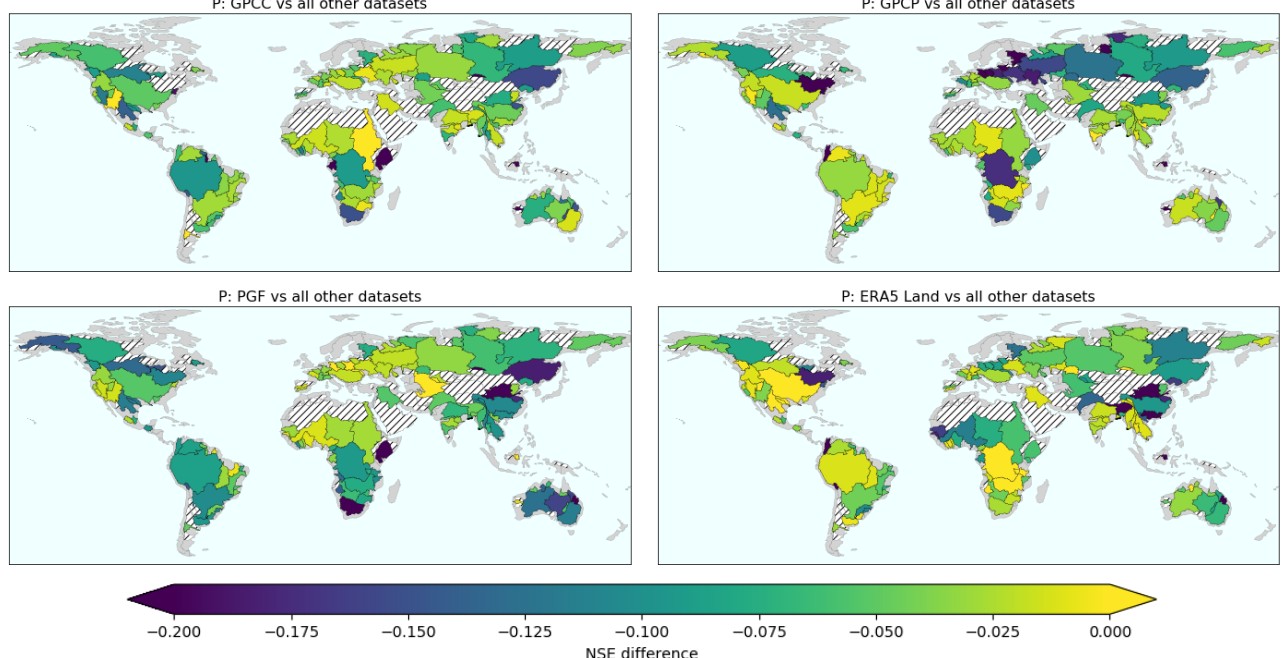

**Figure 13.** The mean of the 10th highest NSE with combinations comprising the reference dataset (*i.e.* GPCC, GPCP, PGF, or ERA5 Land) is compared to the mean of the 10th highest NSE excluding the reference dataset. Yellow indicates basins where the reference dataset is similar to or better than other precipitation datasets while blues show regions where it was significantly worse. Hatches show basins with a poor water budget closure (maximum NSE lower than 0.8 and maximum $NSE_c$ lower than 0.1).

When examining specific evapotranspiration datasets (FLUXCOM, GLEAM, MOD16, and SSEBop), it appeared that GLEAM led to almost optimal NSE values in Africa and Europe (Fig. A9). We also compared the newly released version 3.5 of GLEAM with the older v3.3 used in this study and found that the new version slightly improved the budget closure in every basin (not shown). FLUXCOM was also consistent in North and South America, Europe, western Russia, and South Asia, though it was outperformed by CLSM and ERA5 Land. Finally, SSEBop and MOD16 brought little improvement to
the water budget closure. The poor performances of MOD16 have already been highlighted by e.g., Pascolini-Campbell et al. (2020) in the CONUS, Bhattarai et al. (2019) in India.

The evaluation of runoff datasets in Fig. 15 confirms the differences exhibited for evapotranspiration (Fig. 14). VIC was mainly suitable in temperate and snow regions even if it performed quite poorly in some snow-dominated basins (Nelson, Saint Lawrence, Pechora, among others) due to overestimation of runoff during summer. It is also clear from Fig. 15 that this
LSM is not well-suited for equatorial and arid basins in South America (except some temperate basins in the extreme South), Africa, Australia, and part of Asia. In those basins, the machine-learning model GRUN was exceptionally good, especially outperforming others in South America. In addition, except in the Amazon basin and China where it has already been said that ERA5 Land was not appropriate, this reanalysis yielded a good runoff estimation.



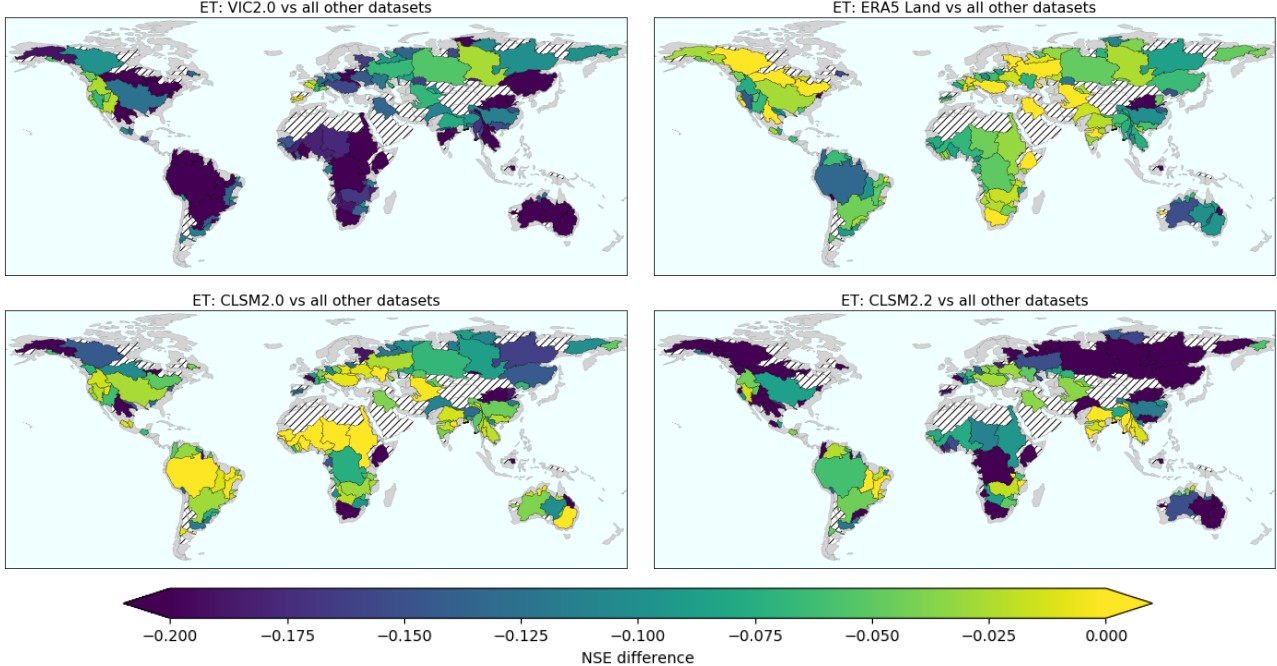

**Figure 14.** Same as 13 but for evapotranspiration datasets.

The low NSE decays on Fig. 15 indicate that the version 2.2 of the Catchment Land Surface Model provide accurate runoff
estimations, which is the main objective of this dataset (Li et al., 2019). However, Fig. 15 shows that it did not improve the
water budget closure achieved by version 2.0 of this same model. In some basins like Congo, the water budget imbalance
increased.

## 5 Conclusions

We assessed the ability of various precipitation, evapotranspiration, and runoff datasets to close the water balance equation
against satellite observed terrestrial water storage anomalies on a global scale. Our analysis was comprehensive as a large
number of global datasets were used to prepare 1694 combinations for closing the water balance in 189 catchments investigated.
We found that the TWSC prediction was better than the long-term mean for 99% of the study area and better than the monthly
mean in 62% of the study area. This illustrates that we can close the water balance equation in most of the regions if we choose
certain datasets for budget components, which is a novel finding in terms of our previous understanding (Lorenz et al., 2014;
Sahoo et al., 2011). We demarcated river catchments where the usual metrics (NSE, cyclostationary NSE) were of limited
interest to evaluate the imbalance error.

Although the lowest imbalance error possible was generally small, we found that none of the 1694 combinations assessed
succeeded in closing the water budget worldwide. Some combinations performed better in regions but underperformed in

**Figure 15.** Same as 13 but for runoff datasets.

others. The combination with all the budget components from reanalysis ERA5 Land was the best in terms of achieving a

positive cyclostationary NSE over the largest fraction of the area under investigation. Individual components (P, ET, and R) of

ERA5 Land were also close to the best performing datasets, except for around the Amazon basin and Eastern China.

The Catchment Land Surface Model additionally appeared as a suitable dataset in many regions excluding snow-dominated

basins. However, version 2.2 of this LSM, which assimilates GRACE data, performed poorly compared to its previous versions.

In some snow-dominated basins, it even led to highly unrealistic ET values during the cold season. Despite being designed

for better runoff estimates, this latest version did not bring much improvement to other runoff datasets in terms of the water

imbalance error. In contrast, GRUN, a machine learning runoff dataset, considerably reduced the imbalance error in several

basins, with the best performances being detected in South America, South Asia, and some Arctic basins in Russia and Alaska.



We have presented a comprehensive overview of our ability to close the global water balance with the help of a wide range of water budget components disseminated for scientific studies. For each water budget component, we also assessed the performance of individual datasets with respect to the other datasets available, which helped us to infer the quality of the dataset when closing the water budget. We also found that the water balance can close due to a cancellation of errors in budget components, therefore, caution should be practiced when closing the water budget over a catchment or region and a large number of datasets should be explored to avoid obtaining right results for wrong reasons. We hope that our analysis will help fellow researchers in finding the most appropriate datasets for water budget analysis in different parts of the world.

*Code availability.* Our code is made available at https://github.com/lehmannfa/water_budget_closure.



**Table 1.** Precipitations datasets

| Name | Method | Period | Spatial resolution | Reference |
|---|---|---|---|---|
| CPC Unified | Rain-gauge | 1979 - present | 0.5° x 0.5° | Chen and Xie (2008) |
| CRU v4.04 | Rain-gauge | 1901 - 2019 | 0.5° x 0.5° | Harris et al. (2020) |
| GPCC v.2020 | Rain-gauge | 1891 - 2019 | 0.5° x 0.5° | Schneider et al. (2020) |
| GPCP v2.3 | Rain-gauge and satellite | 1979 - present | 2.5° x 2.5° | Adler et al. (2018) |
| GPM IMERG v06 | Satellite | 2000 - present | 0.1° x 0.1° | Huffman et al. (2019) |
| TRMM (TMPA/3B43) | Satellite | 1998 - 2019 | 0.25° x 0.25° | Huffman et al. (2007, 2010) |
| ERA5 Land | Reanalysis | 1981 - present | 0.1° x 0.1° | Muñoz-Sabater (2019) |
| JRA55 | Reanalysis | 1958 - present | ~0.5° x 0.5° | Kobayashi et al. (2015); Harada et al. (2016) |
| MERRA2 | Reanalysis | 1980 - present | 0.5° x 0.625° | Reichle et al. (2017) |
| PGF | Rain-gauge, Satellite, and Reanalyses | 1948 - 2014 | 1.0° x 1.0° | Sheffield et al. (2006) |
| MSWEP v2.8 | Rain-gauge, Satellite, and Reanalyses | 1979 - present | 0.1° x 0.1° | Beck et al. (2019) |





**Table 2.** Evapotranspiration datasets

| Name | Method | Period | Spatial resolution | Reference |
|---|---|---|---|---|
| GLDAS2.0 CLSM2.5 | Land surface model | 1948 - 2014 | 1.0° x 1.0° | Li et al. (2020); Koster et al. (2000) |
| GLDAS2.1 CLSM2.5 | Land surface model | 2000 - 2020 | 1.0° x 1.0° | Li et al. (2020); Koster et al. (2000) |
| GLDAS2.2 CLSM2.5 | Land surface model | 2003 - 2020 | 0.25° x 0.25° | Li et al. (2019, 2020) |
| GLDAS2.0 NOAH3.6 | Land surface model | 1948 - 2014 | 1.0° x 1.0° | Beaudoing et al. (2019); Chen et al. (1996) |
| GLDAS2.1 NOAH3.6 | Land surface model | 2000 - 2020 | 1.0° x 1.0° | Beaudoing et al. (2019); Chen et al. (1996) |
| GLDAS2.0 VIC4.1.2 | Land surface model | 1948 - 2014 | 1.0° x 1.0° | Beaudoing et al. (2020); Liang et al. (1994) |
| GLDAS2.1 VIC4.1.2 | Land surface model | 2000 - 2020 | 1.0° x 1.0° | Beaudoing et al. (2020); Liang et al. (1994) |
| FLUXCOM | Machine learning (remote sensing only) | 2001 - 2015 | 0.5° x 0.5° | Jung et al. (2019) |
| GLEAM v3.3a | Priestley-Taylor | 1980 - 2018 | 0.25° x 0.25° | Martens et al. (2017); Miralles et al. (2011) |
| MOD16 | Penman-Monteith | 2000-2015 | 0.5° x 0.5° | Mu et al. (2011) |
| SSEBop | Surface energy balance | 2003 - 2020 | 0.5° x 0.5° | Senay et al. (2013) |
| ERA5 Land | Reanalysis (Penman-Monteith) | 1981 - present | 0.1° x 0.1° | Muñoz-Sabater (2019) |
| JRA55 | Reanalysis (JMA Simple Biosphere SiB) | 1958 - present | 0.5° x 0.5° | Kobayashi et al. (2015); Harada et al. (2016) |
| MERRA2 | Reanalysis (Penman-Monteith) | 1980 - present | 0.5° x 0.625° | Gelaro et al. (2017) |





**Table 3.** Combinations with the largest area covered with a positive cyclostationary NSE

|  | total area with $NSE_c > 0$ (in million $km^2$) | total area with $NSE > 0$ (in million $km^2$) |
|---|---|---|
| *P: ERA5 Land ; ET: ERA5 Land ; R: ERA5 Land* | 35.5 | 89.7 |
| *P: PGF ; ET: CLSM2.0 ; R: CLSM2.0* | 30.8 | 90.2 |
| P: ERA5 Land ; ET: ERA5 Land ; R: CLSM2.2 | 24.5 | 79.7 |
| P: PGF ; ET: NOAH2.0 ; R: CLSM2.0 | 23.9 | 90.9 |
| *P: GPCP ; ET: CLSM2.1 ; R: CLSM2.1* | 23.4 | 79.2 |
| P: ERA5 Land ; ET: ERA5 Land ; R: GRUN | 22.7 | 81.3 |
| P: MSWEP ; ET: CLSM2.0 ; R: CLSM2.0 | 21.8 | 78.5 |
| P: ERA5 Land ; ET: ERA5 Land ; R: CLSM2.0 | 21.7 | 78.6 |
| P: ERA5 Land ; ET: ERA5 Land ; R: MERRA2 | 21.7 | 76.6 |
| P: GPM ; ET: CLSM2.1 ; R: CLSM2.1 | 21.1 | 80.1 |
| P: GPCP ; ET: CLSM2.1 ; R: CLSM2.0 | 20.8 | 78.4 |
| P: GPCC ; ET: CLSM2.0 ; R: CLSM2.0 | 20.4 | 79.4 |
| P: ERA5 Land ; ET: ERA5 Land ; R: NOAH2.0 | 19.8 | 84.4 |
| P: GPM ; ET: CLSM2.1 ; R: CLSM2.0 | 19.0 | 79.4 |
| *P: MERRA2 ; ET: MERRA2 ; R: MERRA2* | 18.8 | 92.1 |
| P: GPM ; ET: NOAH2.1 ; R: NOAH2.0 | 18.8 | 81.0 |
| P: GPM ; ET: CLSM2.1 ; R: CLSM2.2 | 18.7 | 71.2 |
| P: GPCP ; ET: CLSM2.1 ; R: CLSM2.2 | 18.5 | 74.6 |
| P: TRMM ; ET: CLSM2.1 ; R: CLSM2.1 | 18.5 | 56.7 |
| P: PGF ; ET: NOAH2.0 ; R: CLSM2.2 | 18.4 | 86.3 |
| ... | ... | ... |
| *P: PGF ; ET: VIC2.0 ; R: VIC2.0* | 16.1 | 87.6 |
| ... | ... | ... |
| *P: PGF ; ET: NOAH2.0 ; R: NOAH2.0* | 16.0 | 92.4 |
| ... | ... | ... |
| *P: GPCP ; ET: NOAH2.1 ; R: NOAH2.1* | 13.3 | 82.6 |
| ... | ... | ... |
| *P: ERA5 Land ; ET: CLSM2.2 ; R: CLSM2.2* | 10.8 | 57.8 |
| ... | ... | ... |
| *P: JRA55 ; ET: JRA55 ; R: JRA55* | 8.7 | 72.2 |
| ... | ... | ... |
| *P: GPCP ; ET: VIC2.1 ; R: VIC2.1* | 7.1 | 75.6 |

Combinations are ranked by decreasing area of basins with a positive cyclostationary NSE. Italics indicate combinations where P, ET, and R are from the same model.



# 1 Additional figures

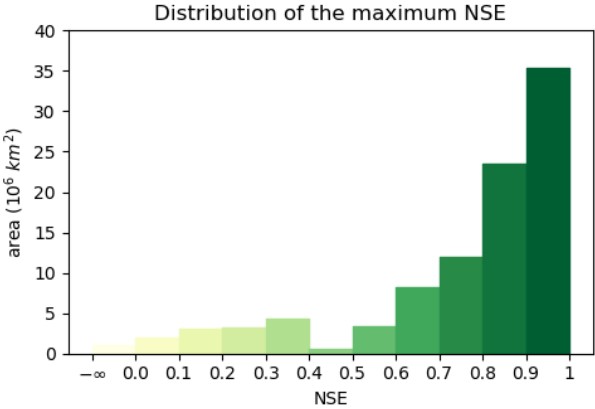

**Figure A1.** Distribution of the maximum NSE over all combinations in terms of basin area



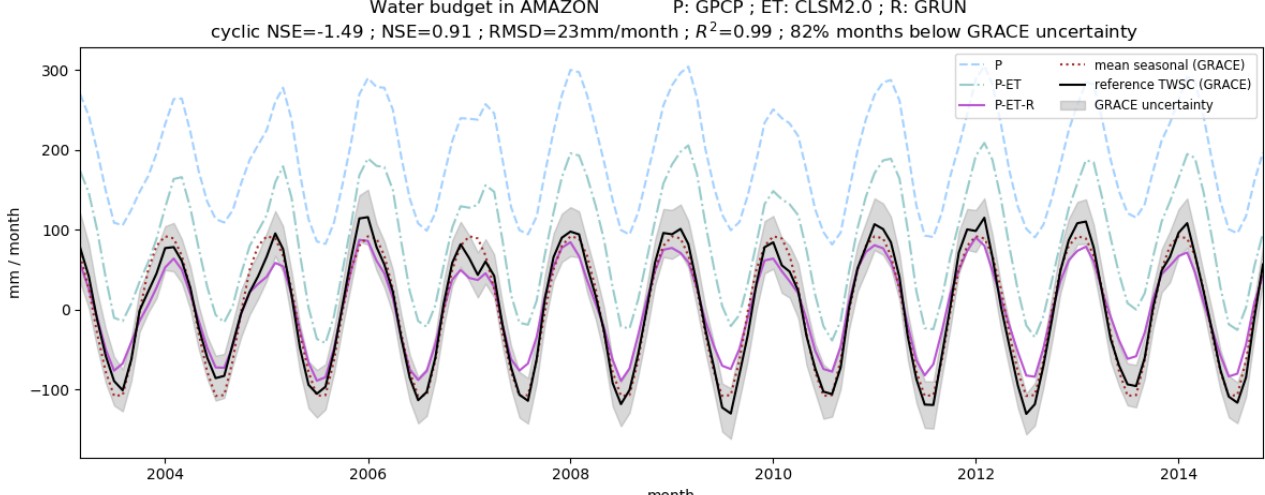

**Figure A2.** Components of the water budget in the Amazon basin for the combination leading to the highest NSE

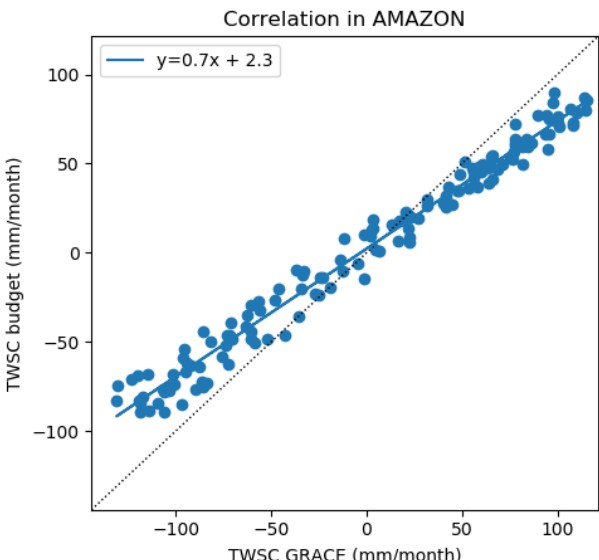

**Figure A3.** Correlation between monthly values of GRACE TWSC and the budget reconstruction in the Amazon basin, with the combination leading to the highest NSE (NSE=0.92 and cyclostationary NSE=-1.28)





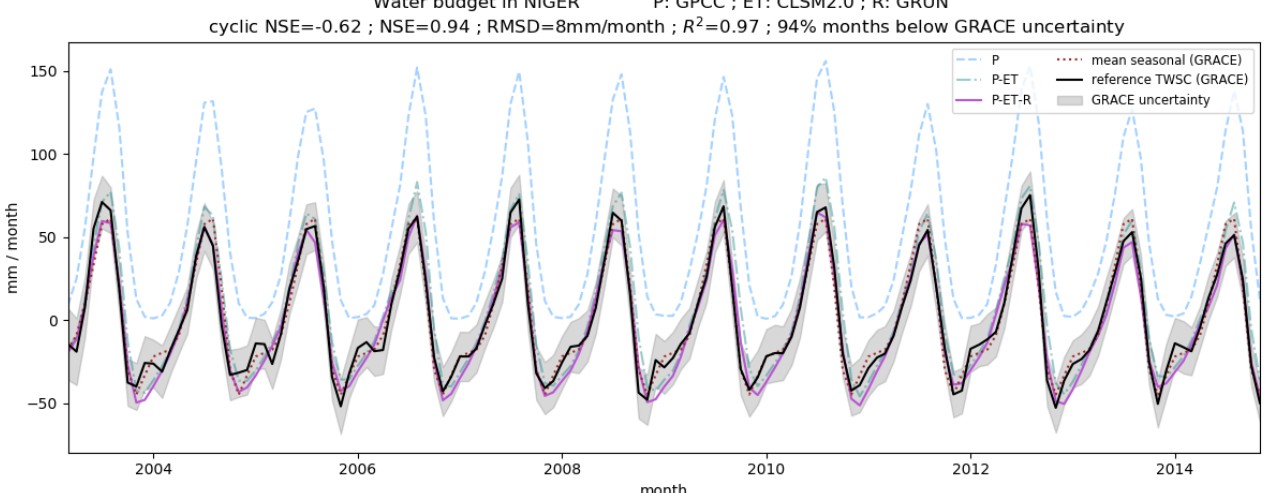

**Figure A4.** Components of the water budget in the Niger basin for the combination leading to the highest NSE



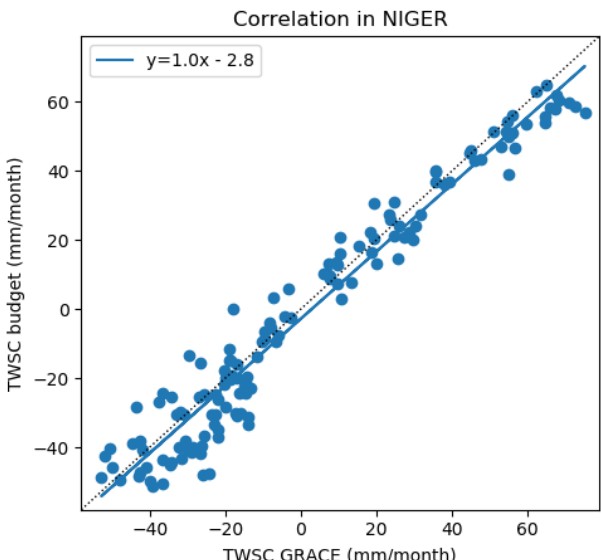

**Figure A5.** Correlation between monthly values of GRACE TWSC and the budget reconstruction in the Niger basin, with the combination leading to the highest NSE (NSE=0.94 and cyclostationary NSE=-0.62)



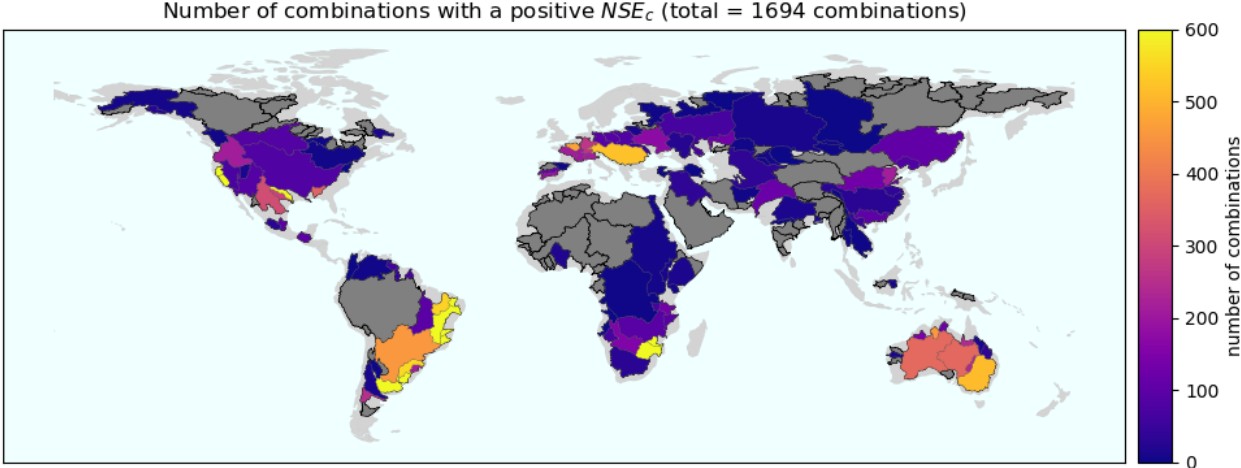

**Figure A6.** Number of combinations yielding a positive cyclostationary NSE in each basin. Grey means that no combination achieved a positive value.



**Figure A7.** 132 basins with a maximum NSE larger than 0.8 or a maximum $NSE_c$ larger than 0.1. The distance between basins is the Euclidean distance between the vector of costs for each combination. The height of the U-shaped link is proportional to this distance. Basins are clustered to minimize the intra-cluster variance and colored basins are those selected to plot Fig. 12





**Figure A8.** The mean of the 10th highest NSE with combinations comprising the reference dataset is compared to the mean of the 10th highest NSE excluding the reference dataset. Yellow indicates basins where the reference dataset is similar to or better than others while blues show regions where it was significantly worse. Hatches show basins with a poor water budget closure (maximum NSE lower than 0.8 and maximum $NSE_c$ lower than 0.1).



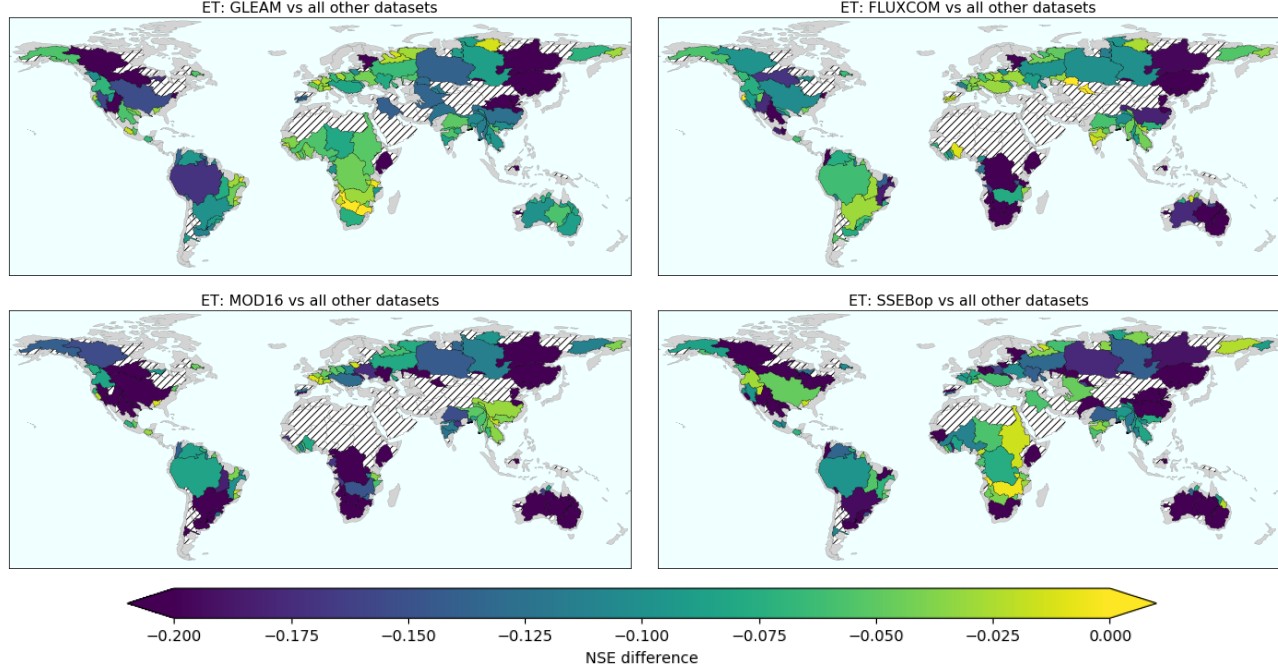

**Figure A9.** The mean of the 10th highest NSE with combinations comprising the reference dataset is compared to the mean of the 10th highest NSE excluding the reference dataset. Yellow indicates basins where the reference dataset is similar to or better than others while blues show regions where it was significantly worse. Hatches show basins with a poor water budget closure (maximum NSE lower than 0.8 and maximum $NSE_c$ lower than 0.1).





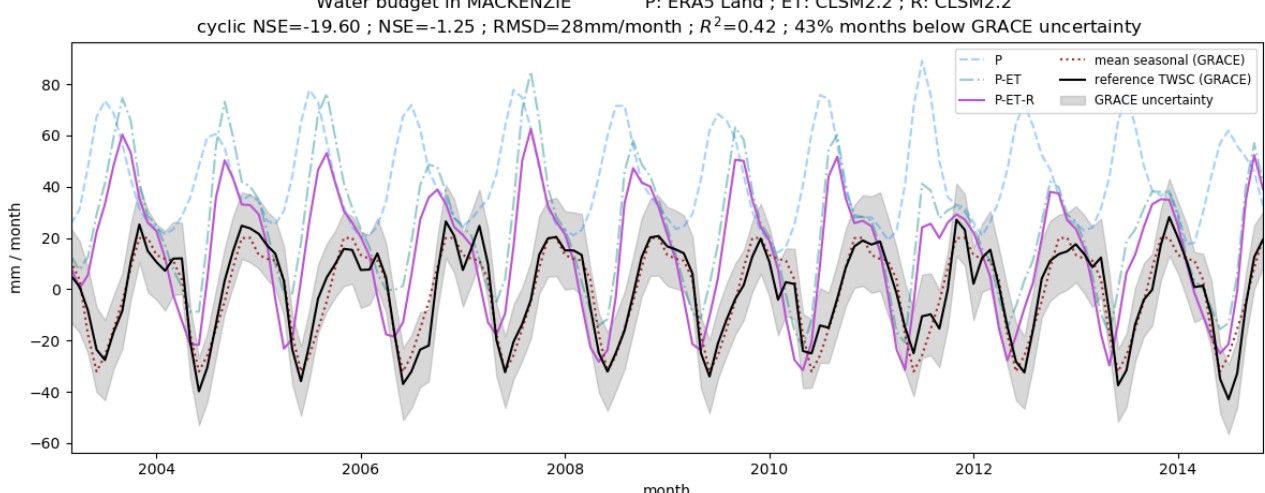

**Figure A10.** Components of the water budget in the Mackenzie basin with all components from GLDAS2.2 CLSM (assimilating GRACE TWS)





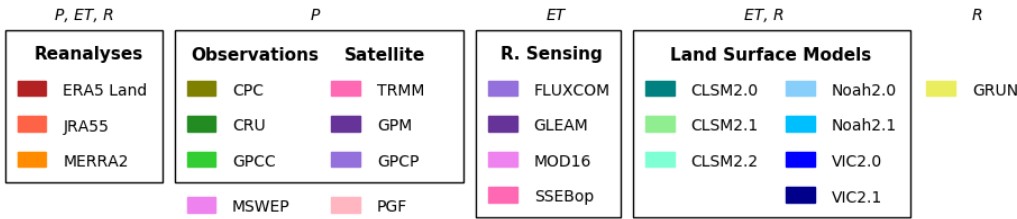

**Figure A11.** Legend of Fig. A12, A13, A14, A15



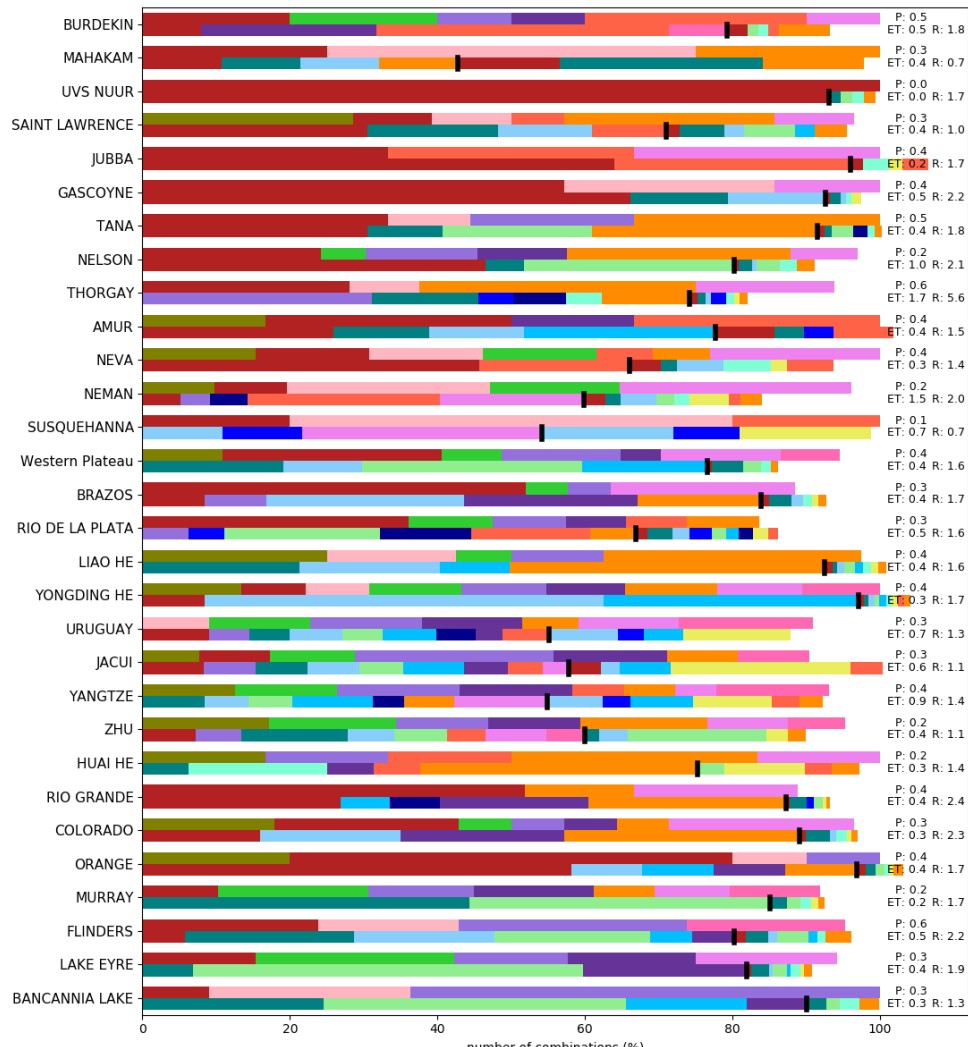

**Figure A12.** Datasets appearing in combinations that satisfy a cost lower than 0.1 for each basin separately. The top line of represents precipitations datasets. The left part of the bottom line is evapotranspiration datasets while the right part is runoff. The limit between ET and R is symbolized by a black line located proportionally to the portion of ET in the mean annual water cycle of the corresponding region, explaining while the bottom line may have a length different than 100%. Basins are ordered according to hierarchical clustering (dendrogram in Fig. A7). The color legend for datasets can be found in Fig. A11





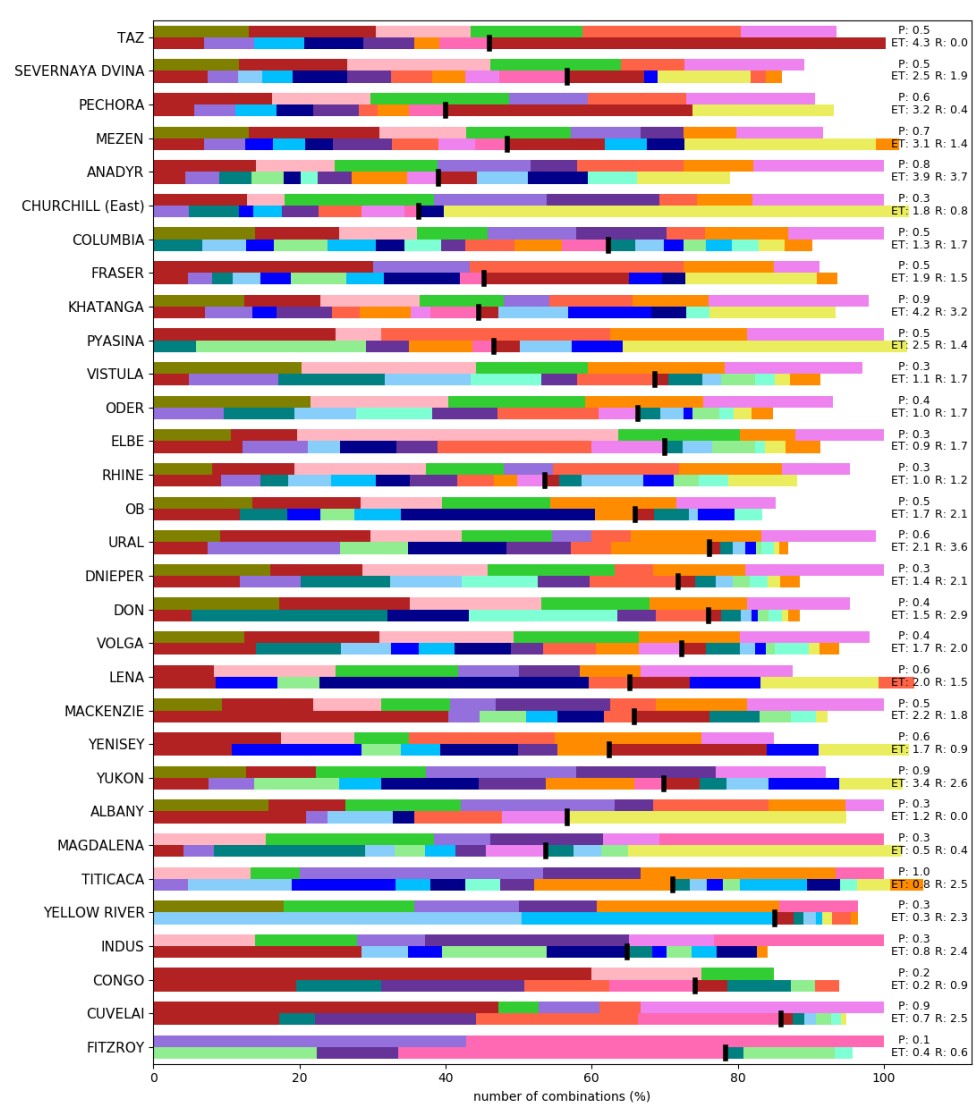

**Figure A13.** Following of Fig. A12







**Figure A14.** Following of Fig. A13


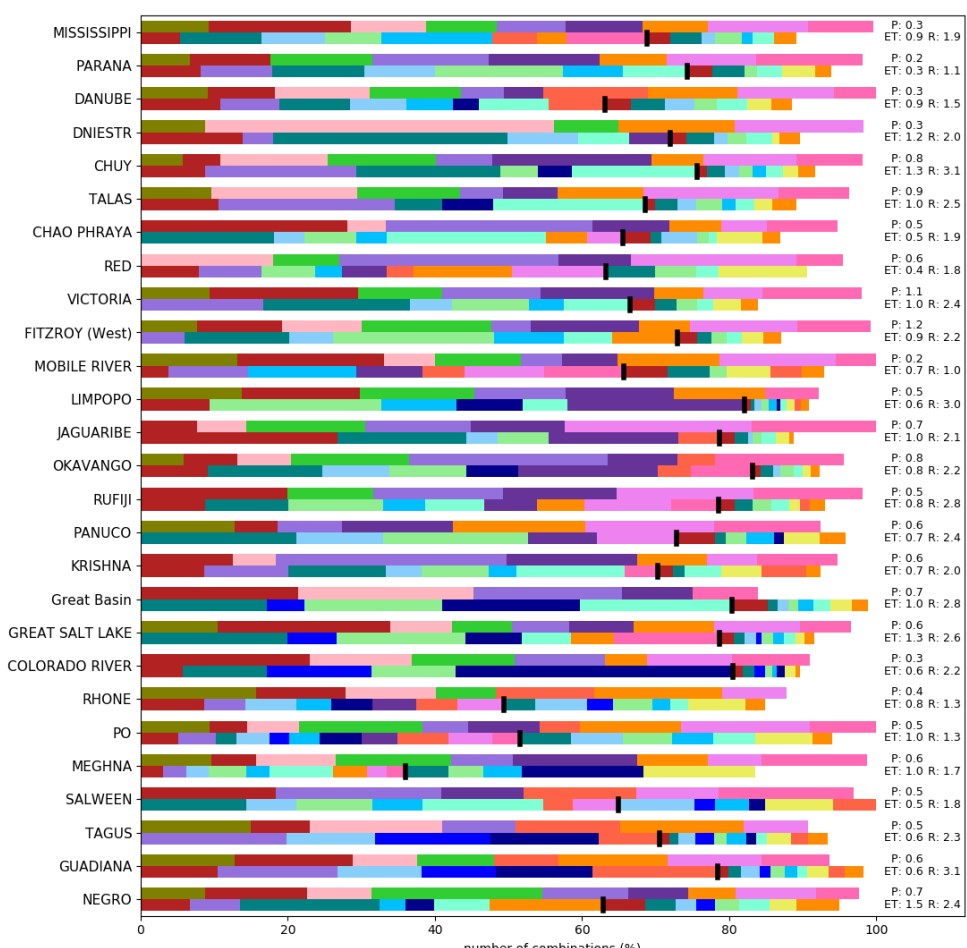

**Figure A15.** Following of Fig. A14



# A1 Additional tables

**Table A1.** Components of the mean annual water cycle in Pacific islands

|  | P (mm/year) | ET (mm/year) | R (mm/year) |
|---|---|---|---|
| SEPIK | 3390 ± 653 | 1404 ± 223 | 2116 ± 597 |
| MAMBERAMO | 3578 ± 851 | 1340 ± 227 | 2406 ± 756 |
| MAHAKAM | 3163 ± 356 | 1359 ± 272 | 1911 ± 529 |
| KAPUAS | 3666 ± 204 | 1366 ± 266 | 2339 ± 480 |

The first value is the mean annual cycle averaged over all datasets while the second one is
the standard deviation of mean annual values over all datasets





**Table A2.** Components of the mean annual water cycle in equatorial rain forest/monsoon basins in South America

|              | P (mm/year)     | ET (mm/year)    | R (mm/year)     |
|--------------|-----------------|-----------------|-----------------|
| MAGDALENA    | 2339 ± 650      | 1157 ± 216      | 1373 ± 498      |
| CUYUNI       | 2051 ± 269      | 1395 ± 223      | 766 ± 327       |
| ESSEQUIBO    | 2121 ± 251      | 1314 ± 217      | 946 ± 405       |
| MARONI       | 2312 ± 247      | 1406 ± 269      | 885 ± 364       |
| AMAZON       | 2177 ± 172      | 1251 ± 196      | 958 ± 251       |
| ORINOCO      | 2269 ± 289      | 1237 ± 200      | 1090 ± 315      |

The first value is the mean annual cycle averaged over all datasets while the second one
is the standard deviation of mean annual values over all datasets





*Author contributions.* J. B. and B. D. V. designed the experiment. F.L. implemented the code and wrote the paper with support from all coauthors. All the authors contributed to the synthesis of results and key conclusions.

*Competing interests.* JLB and FL were supported by the European Research Council (ERC) under the European Union's Horizon 2020 research and innovation programme under grant agreement No 694188 (GlobalMass) and BDV is supported by the Marie Skłodowska-Curie Individual Fellowship (MSCA-IF) under grant agreement no 841407 (CLOSeR).

*Acknowledgements.* We are grateful to Miss Megan Rounsley, who carefully proofread the manuscript. Concerning the datasets used, they are all publicly available and the links to download them can be found in our Github repository. CPC Global Unified Precipitation data
provided by the NOAA/OAR/ESRL PSL, Boulder, Colorado, USA, from their Web site at https://www.psl.noaa.gov/data/gridded/data.cpc. globalprecip.html. GPCP data provided by the NOAA/OAR/ESRL PSL, Boulder, Colorado, USA, from their Web site at https://psl.noaa.gov/ data/gridded/data.gpcp.html. CSR mascons were downloaded from http://www2.csr.utexas.edu/grace and GRACE/GRACE-FO JPL Mascon data are available at http://grace.jpl.nasa.gov.



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
