# Peer review of "How well are we able to close the water budget at the global scale?"

_Hydrology and Earth System Sciences, 2021_

## Referee Comment (RC1)

**Review comments for the manuscript** "How well are we able to close the water budget at the global scale?" by Fanny Lehmann, Bramha Dutt Vishwakarma, and Jonathan Bamber.

**General**

The aim of this paper is to provide a revised overview of the water budget closure on a global scale by evaluating 1694 combinations of P-ET-R. Detailed comments are given below.

**Specific Concerns/Comments**

1) Pages 8 and 9: The author mentioned that they applied the method of Long et al. (2014b) to estimate TWSC. However, the formula of Long et al. (2014b) shown below was different from that used in this manuscript. Please double check.

ble in Eq. (1). GRACE TWSC is computed as the backwards difference of TWSA (mm) whose reference is the mean gravity field for a calculation period (e.g., Jan 2003–Sep 2012 in this study):

$$dS/dt = \frac{\text{TWSA}(t) - \text{TWSA}(t-1)}{t}. \tag{2}$$

2) Page 9: It was stated that time-series of P, ET, and R also needed to be time-filtered by equation 3. In my opinion, P, ET, and R are fluxes, while TWS is a state variable. For one given month, P represents the total precipitation that occurs in that month. The use of equation 2 was likely due to that GRACE provides noisy monthly TWSA. It is kind of difficult for me to understand time-filtered P, ET and R. Please clarify.

3) It was concluded that TWS changes reconstructed from the water balance equation (P-ET-R) were more accurate than the long-term and monthly mean of GRACE time series in the corresponding basins. Was the GRACE-TWSC used as benchmark data in the manuscript? If so, I was confused by the conclusion that P-ET-R was more accurate than GRACE-TWSC. I may not understand it. Could the author explain it more?

4) Figure A12: "The top line of represents precipitations datasets." I did not get it.

5) Page 24: "However, version 2.2 of this LSM, which assimilates GRACE data, performed poorly compared to its previous versions." There was an explanation: "Since this dataset assimilates GRACE measurements and was validated against GRDC observations, this may reflect overfitting of runoff that is better constrained than evapotranspiration, therefore leading to unrealistic ET values." Could the author explain it more? Generally, the following seems more easy to understand: when runoff became more accurate, ET would be improved accordingly.

6) Typo mistakes:

Pages 1, 21 and 23: The "Catchment Land Surface Model" was supposed to be CLSM as the abbreviation had appeared earlier.

"Fig. 2" and "Figure 2" were both used in the manuscript. Please unify the expression.

---

## Referee Comment (RC2)

**Review of HESS-2021-279 - How well are we able to close the water budget at the global scale?**

In their study, the authors present a comprehensive analysis of the water budget closure over a large number of river basins using a wide range of state-of-the-art global hydrometeorological data. While I have several major and minor comments, I found the manuscript very well written and clearly structured. As the study therefore presents a very comprehensive and detailed overview of current global hydrometeorological datasets and, hence, is a good starting point for future research e.g. on a more regional scale, I suggest a publication after major revisions (see especially my major comments 1 and 2).

**Major comments:**

- Using the GURN-dataset as the only source of information for runoff is a bit disappointing. It is certainly correct that LSMs without any routing scheme might provide runoff estimates that are not directly comparable to observed runoff on daily or shorter time-scales. But as the authors are analysing monthly and longer-term averages (and even apply a temporal filter), the inconsistencies between observed and modelled runoff should substantially decrease. Moreover, the reasons why we need such comprehensive evaluations and why we see such large discrepancies in the water budgets are that we do not have global reliable information about catchment-scale precipitation and evapotranspiration. But for gauged basins, runoff is usually the most accurately observed variable in the water balance equation. The authors claim that "it is useful to know beforehand which datasets are more reliable to close the water budget in the region under study" (page 11, line 250). And we do have this information for a large number of basins that are analysed in this study. While GURN is certainly a valuable dataset, it is again one step away from the "real" observed quantity, which we, in the end, try to reproduce with all our different global models and remote sensing products.

  Another issue with the GURN dataset is that it is only valid for rivers without extensive human interactions (reservoir management, substantial extraction for irrigation, etc.). This is mentioned by Ghiggi et al. (2019) themselves in chapter 5.5. But using TWSCs from GRACE (as in this study) actually allows to take such interactions into account as e.g. retention of water in reservoirs is reflected in the basin-scale total water storage. Thus, using datasets like GURN as a proxy for observed runoff can actually make results worse than they actually are.

  Thus, if possible, it would make the study even more comprehensive and convincing if the authors also include a similar evaluation with observed runoff at least for a subset of study basins.

- I find the separation into Köppen-Geiger-classes a bit problematic. Many river basins extend over multiple climate-zones and -classes. Moreover, the climate zone of the headwaters (i.e. where most of the runoff is generated) might be much more important for the characterisation of the basins than the downstream areas. So while it certainly makes sense to somehow categorise and cluster the basins and, hence, try to better understand the water budget closure, I assume that a different "metric" (e.g. drainage area, length of the rivers, relationship between rainfall and runoff, sources of moisture, catchment-scale characteristics (topography, gradients, etc.), advection vs. local evapotranspiration) could give even more insight into the performance of current hydrometeorological datasets. See e.g. https://hydrosheds.org/page/hydroatlas for a quite new approach for characterising river basins. If possible, it would be worth to check if some of these metrics give a clearer picture of the regional performance.

- While I really acknowledge the huge amount of work that the authors have put in their study, I think that the manuscript is only a starting point for future research. The authors claim that a positive NSE could be achieved over 99% of the basins but only if we choose the right combination of datasets. Thus, for an area of 35.5 mio. km^2 (according Table 3), ERA5 Land might not be the most realistic dataset but best dataset in terms of bias cancellation. The reasons for this remain (mostly) unknown. This, however, is not the authors fault but somehow due to the very nature of such global assessments. Future studies and regional applications must therefore use the findings from this study (e.g. from figures A12-A15) as a starting point to further explore strengths and weaknesses of individual datasets across different regions. Only then are we able to see improvements in our global hydrometeorolgocal data sources (as the authors also state in the abstract).

**Minor comments:**

- The paper is very long and contains a huge amount of quite detailed information! I would hence really urge the authors to reduce the number of pages! As a suggestion, they could put the whole description of methods (water budgets, central differences, filtering of hydrometeorological information, RMSD, NSE, etc.) into the supplementary material as these topics have been presented in many other studies.
- The authors mention that they bring every dataset to a resolution of 0.5°. This, however, could lead to issues for coarser datasets (GLDAS, GPCP, etc.), particularly for smaller catchments. Are there any relationships between the resolution of the input datasets and the performance metrics?
- Page 10, line 223: I guess $err\_cst$ are simply anomalies, right? Then, $err\_cst^2$ is simply the temporal variability of total water storage changes. If this is the case, I would not call these *errors* as this sounds misleading.
- Page 10, equation 6: Similarly, $err\_cyc^2$ is just the deviation from the annual cycle and, again, using the term *error* for such anomalies is quite misleading.
- Page 11, line 260: The two consecutive enumerations (i.e. lines 257 - 259 and 260 onwards) look a bit weird... Please add at least one sentence for separating these two parts.
- page 12, line 263: ...is within the confidence interval from GRACE TWSCs.
- Page 13, lines 294 - 295: Do you have an explanation why the performance has improved? Is it due to improvements of the consistency or the performance of the hydrometeorological datasets?
- Page 14, lines 315 - 325: Do you have any explanation why TWSC is too low in the wet and too high during the dry season, respectively? According to Fig. A3, this under- and overestimation seems to be quite systematic.
- Page 15, line 350: Important for what?
- Page 18: line 398: For this analysis, we focus on a subset of 132 basins out of the 189, where an excellent budget closure could be achieved.
- Page 19, line 405: This is a dangerous conclusion as it indicates that two very "bad" datasets can still lead to good water budget closure, if there occurs a cancellation of biases (i.e. right for the wrong reason), right? This would mean that e.g. the datasets in Figure 12, that satisfy a cost lower than 0.1, must not necessarily be realistic datasets but, by combining them with other suitable datasets, only achieve a reasonable water budget closure.
- Page 19, lines 12 - 16 and Figure A7: I did not really understand the clustering approach. What exactly are the authors trying to do here? Do they want to define 13 representative catchments and then identify smaller basins that achieve a similar performance? If this is the case, why did they choose 13 "artificial" clusters instead of using e.g. similar climatic or topographic conditions?

- Page 19, line 430: This is an important statement as GPCP has by far the lowest spatial resolution of 2.5° (around 250km). Claiming that GPCP (approx. 250km) performs similar than ERA5 Land (9km) indicates that resolution does not play a big role, even this is generally assumed by the community (especially over complex terrain). Could you find any relationship between the performance of GPCP and the size of the catchments?
- Page 21, lines 450 - 455: The authors suggest that the discrepancies between the TWSC from water budgets and GRACE are somehow related to overfitting of the CLSM. But there is also a huge temporal shift between the two time-series. Are there any explanations for this?
- Page 21, Figure 12 (and A12-A15).: Is there any meaning of the length of an individual section (or dataset)? And at least on long-term averages, we often assume that P should be equal to ET + R but this is not the case for several clusters. Can you explain why this happens here? Or did I misinterpret the figures?
- Page 22, Figure 13: As the distribution of NSE-values might be highly skewed, wouldn't it make more sense to show the median of the 10 best-performing combinations?
- Pages 21 and 39, Captions for Figures 12 and A11: Why are MSWEP, PGF and GRUN outside the boxes

---

## Author Comment (AC1)

How well are we able to close the water budget at the global scale?, F. Lehmann, B. D. Vishwakarma, J. Bamber

**Reply to referee 1**

We thank Christof Lorenz for his detailed comments on our manuscript. Our reply is attached below.

*Major comments*

*Using the GURN-dataset as the only source of information for runoff is a bit disappointing. […] , if possible, it would make the study even more comprehensive and convincing if the authors also include a similar evaluation with observed runoff at least for a subset of study basins.*

The reason for using GRUN was to have temporally consistent time-series for all basins. However, we agree that it does not represent real variations. We selected basins with a discharge area similar to the boundaries of the basins we studied and computed the runoff time-series from GRDC measurements. We combined these time-series with P and ET data to obtain 154 combinations. The table below shows that including runoff measurements improved the water budget closure in this selection of basins.  The maximum NSE and cyclostationary NSE were computed over the 154 combinations containing GRDC measurements. The values were compared with the maximum NSE and cyclostationary NSE over the 1694 combinations used in the manuscript (with R from models).

| | max NSE | max NSE GRDC | max NSEc | max NSEc GRDC | months with available runoff data (%) |
|---|---|---|---|---|---|
| **AMAZON** | 0.907397 | 0.980957 | -1.490078 | 0.487929 | 100.0 |
| **AMUR** | 0.586040 | 0.898816 | 0.459075 | 0.867783 | 31.9 |
| **CONGO** | 0.860302 | 0.867065 | 0.178806 | 0.218559 | 66.0 |
| **DANUBE** | 0.903778 | NaN | 0.628426 | NaN | 0.0 |
| **LENA** | 0.825520 | NaN | -0.229911 | NaN | 0.0 |
| **MACKENZIE** | 0.889386 | 0.911745 | -0.010903 | 0.193439 | 100.0 |
| **MISSISSIPPI** | 0.932459 | 0.940384 | 0.469833 | 0.532042 | 84.4 |
| **OB** | 0.915694 | 0.957197 | 0.227638 | 0.607863 | 66.0 |
| **ORANGE** | 0.341260 | 0.205192 | 0.196090 | 0.030036 | 100.0 |
| **PARANA** | 0.902606 | 0.899179 | 0.688144 | 0.677171 | 95.0 |
| **VOLGA** | 0.918098 | 0.937325 | 0.445893 | 0.575976 | 66.0 |
| **YANGTZE** | 0.747095 | NaN | 0.214342 | NaN | 7.1 |
| **YELLOW RIVER** | 0.740573 | NaN | 0.501178 | NaN | 7.1 |
| **YENISEY** | 0.920923 | 0.935921 | 0.269030 | 0.407666 | 74.5 |
| **YUKON** | 0.904029 | 0.931050 | 0.106887 | 0.358348 | 100.0 |

*I find the separation into Köppen-Geiger-classes a bit problematic. […] If possible, it would be worth to check if some of these metrics give a clearer picture of the regional performance.*

With the Köppen-Geiger classification, we tried to understand which factors could explain similarities between basins. It is not perfect but one of its advantages is that its assumptions are well-known. We have already explored some variables exposed in the HydroSHEDS database (annual discharge, lake volume, reservoir volume, degree of regulation) and we did not find any significant relationship between those variables and the imbalance error.

We also tried a clustering method based on the shape of the TWS seasonal signal that did not help further. Below is an example of the four clusters detected by the algorithm (K-means clustering) with the mean seasonal signal representative of each cluster. However, the clusters were highly dependent on the initialization of the algorithm and we were not able to understand why a basin belonged to the selected cluster. Therefore, this idea was not explored further.

[Figure]

*Basins clustered in 4 groups using a K-means algorithm based on the shape on the mean annual cycle of TWS*

[Figure]

*Mean annual cycle of TWS representative of each cluster*

Concerning the fact that a basin may be comprised of several climate zones, we specifically studied the Amazon sub-basins that are of smaller sizes and less diverse climates. The best combinations in the sub-basins were similar to those found for the entire Amazon basin, leading us to think that aggregating basins with some heterogeneous climates was not a major drawback.

*[…] Future studies and regional applications must therefore use the findings from this study (e.g. from figures A12-A15) as a starting point to further explore strengths and weaknesses of individual datasets across different regions. Only then are we able to see improvements in our global hydrometeorological data sources (as the authors also state in the abstract).*
We entirely agree that our study is only a starting point. It highlights a need for independent validation data that we hope to develop in the future.

*Minor comments*
*The paper is very long and contains a huge amount of quite detailed information! I would hence really urge the authors to reduce the number of pages!*

As suggested, we have reduced the length of the article and clarified which material belongs to the supplementary information.

***The authors mention that they bring every dataset to a resolution of 0.5°. This, however, could lead to issues for coarser datasets (GLDAS, GPCP, etc.), particularly for smaller catchments. Are there any relationships between the resolution of the input datasets and the performance metrics?***
In this study we have explored more than 1600 combinations. Having the same resolution for all datasets was necessary to ensure the consistency of the computations that included numerour loops. Since we are only interested in basins above the GRACE spatial resolution (>65000 sq. km), we believe that changing resolution should not have a significant impact on basin-mean time-series on a monthly time scale; and we did not notice any result that could suggest it might be the case. A more detailed reply is given below comparing GPCP and ERA 5 Land.

***Page 10, line 223: I guess err_cst are simply anomalies, right? Then, err_cst^2 is simply the temporal variability of total water storage changes. If this is the case, I would not call these errors as this sounds misleading.***
***Page 10, equation 6: Similarly, err_cyc^2 is just the deviation from the annual cycle and, again, using the term error for such anomalies is quite misleading.***
It is right that the terms err_cst and err_cyc represent deviations and therefore may be misleading. We have modified those names.

***Page 11, line 260: The two consecutive enumerations (i.e. lines 257 - 259 and 260 onwards) look a bit weird... Please add at least one sentence for separating these two parts.***
***page 12, line 263: ...is within the confidence interval from GRACE TWSCs.***
The suggested corrections have been made.

***Page 13, lines 294 - 295: Do you have an explanation why the performance has improved? Is it due to improvements of the consistency or the performance of the hydrometeorological datasets?***
The main improvements come from the reference variable used to compute the NSE. Since R is generally small compared to P and ET, the NSE is higher with TWS as the reference rather than with R. The two figures below illustrate this with a combination found in [Lorenz et al., 2014].
The top figure shows TWS as the reference and leads to a high NSE (0.69) whereas the bottom one has R as a reference and shows a negative NSE, similarly to [Lorenz et al., 2014].

[Figure]

[Figure]

**Page 14, lines 315 - 325: Do you have any explanation why TWSC is too low in the wet and too high during the dry season, respectively? According to Fig. A3, this under- and overestimation seems to be quite systematic.**

In the Amazon basin, the under- and over- estimation was coming from inappropriate values of R computed by the GRUN dataset. Top figure: with GRUN, bottom: with GRDC gauge measurements. However, we also note that the basin used by the GRDC is 20% smaller than ours.

[Figure]

The sentences have been rephrased as suggested.

**Page 19, line 405: This is a dangerous conclusion as it indicates that two very "bad" datasets can still lead to good water budget closure, if there occurs a cancellation of biases (i.e. right for the wrong reason), right? This would mean that e.g. the datasets in Figure 12, that satisfy a cost lower**

*than 0.1, must not necessarily be realistic datasets but, by combining them with other suitable datasets, only achieve a reasonable water budget closure.*

When examining individual datasets, the risk of bias cancellation seems unavoidable since we cannot *a priori* exclude some datasets. This is why we implement the clustering approach (see the reply to the next question).

*Page 19, lines 12 - 16 and Figure A7: I did not really understand the clustering approach. What exactly are the authors trying to do here? Do they want to define 13 representative catchments and then identify smaller basins that achieve a similar performance? If this is the case, why did they choose 13 "artificial" clusters instead of using e.g. similar climatic or topographic conditions?*

When examining datasets leading to a small imbalance error in a given basin, it appeared that many dataset combinations performed equally well. However, they would frequently lead to a low performance in a neighbouring basin with seemingly similar conditions (size, climate zone, or shape of the mean seasonal TWS). This prevented us from drawing any conclusion on the ability of this combination to effectively close the water budget.

Therefore, we tried to increase the constraints on the combinations from closing the water budget in a single basin, to closing the water budget in several basins. Since no combination was able to meet this condition in every basin of a climate zone, we relaxed the constraint to closing the water budget in several basins sharing some similar patterns that we do not necessarily understand. We then used the clustering approach to find groups of basins following the same patterns, i.e. satisfying a low imbalance error for similar combinations.

*Page 19, line 430: This is an important statement as GPCP has by far the lowest spatial resolution of 2.5° (around 250km). Claiming that GPCP (approx. 250km) performs similar than ERA5 Land (9km) indicates that resolution does not play a big role, even this is generally assumed by the community (especially over complex terrain). Could you find any relationship between the performance of GPCP and the size of the catchments?*

Assessing the performances of the datasets with the NSE difference, we found no correlation between the size of the catchments and the performances of GPCP ($R^2$=0.09). There is no significant difference with ERA5 Land ($R^2$=0.06).

[Figure]

*Page 21, lines 450 - 455: The authors suggest that the discrepancies between the TWSC from water budgets and GRACE are somehow related to overfitting of the CLSM. But there is also a huge temporal shift between the two time-series. Are there any explanations for this?*

We realized that our interpretation was maybe too superficial. Therefore, we deleted our explanation and instead suggested that this remains an open question for the community.

***Page 21, Figure 12 (and A12-A15).: Is there any meaning of the length of an individual section (or dataset)? And at least on long-term averages, we often assume that P should be equal to ET + R but this is not the case for several clusters. Can you explain why this happens here? Or did I misinterpret the figures?***

The length of a section represents the proportion of combinations in which each dataset was found. When one dataset has a much longer section than the others, it means that it was able to close the water budget when being combined with several datasets. Hence, this reduces the risk of error cancellation with this dataset. We will clarify this legend.

The total length of the P, ET, and R sections were computed using the annual mean over all datasets. Therefore, we cannot expect that different datasets lead to P=ET+R. Since this is only for representative purposes (to express the x-axis as a percentage and make basins comparable), it should not be misleading.

***Page 22, Figure 13: As the distribution of NSE-values might be highly skewed, wouldn't it make more sense to show the median of the 10 best-performing combinations?***

As shown below, there is very little difference between using the mean or the median since it is computed over the 10 best-performing combinations over 1600 combinations. The possible skewness would appear with a larger number of combinations.

[Figure]

Using the median of the 10th highest combinations

[Figure]

Using the mean (as done in the article)

***Pages 21 and 39, Captions for Figures 12 and A11: Why are MSWEP, PGF and GRUN outside the boxes?***

Datasets were gathered by "type". Since MSWEP and PGF are built using a combination of several methods, they do not really belong to any category among "rain-gauge", "satellite", or "reanalysis". Similarly, GRUN is the only machine learning dataset for runoff and therefore was separated from land surface models.

---

## Author Comment (AC2)

How well are we able to close the water budget at the global scale?, F. Lehmann, B. D. Vishwakarma, J. Bamber

**Reply to referee 1**

We thank the reviewer for his/her time in reading our manuscript. A detailed reply to the comments made can be found below.

*1) Pages 8 and 9: The author mentioned that they applied the method of Long et al. (2014b) to estimate TWSC. However, the formula of Long et al. (2014b) shown below was different from that used in this manuscript. Please double check.*

This was a referencing error; the reference should have been Long et al. (2014a).

*2) Page 9: It was stated that time-series of P, ET, and R also needed to be time-filtered by equation 3. In my opinion, P, ET, and R are fluxes, while TWS is a state variable. For one given month, P represents the total precipitation that occurs in that month. The use of equation 2 was likely due to that GRACE provides noisy monthly TWSA. It is kind of difficult for me to understand time-filtered P, ET and R. Please clarify.*

TWS is indeed a state variable that describes the amount of water at a certain time $t$. The difference between two months can be written as the integral of the accumulation rate $A(t) = P(t) - ET(t) - R(t)$,

$$TWS(t+1) - TWS(t-1) = \int_{t-1}^{t+1} A(\tau)d\tau$$

The integral can be either approximated by the value at its center, in which case the equation becomes $TWS(t+1) - TWS(t-1) = A(t) * (t+1-(t-1)) \approx 2\Delta t \times A(t)$

However, it is known that an integral is better approximated using the trapezoidal rule, which gives

$$TWS(t+1) - TWS(t-1) = \frac{\Delta t}{2}\left(A(t-1) + A(t)\right) + \frac{\Delta t}{2}\left(A(t) + A(t+1)\right)$$

The last formula corresponds to the time filter described in equation (3). On the one hand, the filter smooths slightly the time-series, which removes some high-frequency variations and may lead to optimistic results. On the other, annual peaks are reduced, which may create some underestimation. [Landerer et al., 2010] found that the imbalance error was higher when not using the time filtering, which also reflects our findings.

Below is an example of the water budget closure in the Mississippi basin with (top figure) and without (bottom figure) time filtering.

[Figure]

[Figure]

**3) It was concluded that TWS changes reconstructed from the water balance equation (P-ET-R) were more accurate than the long-term and monthly mean of GRACE time series in the corresponding basins. Was the GRACE-TWSC used as benchmark data in the manuscript? If so, I was confused by the conclusion that P-ET-R was more accurate than GRACE-TWSC. I may not understand it. Could the author explain it more?**

GRACE TWSC was used as benchmark data but there might have been a misunderstanding. The conclusion that P-ET-R was more accurate than GRACE TWSC refers only to the long-term or annual mean. This comes from the interpretation of the NSE and cyclostationary NSE. They indicate that in a situation where TWSC would be unknown, it would be better to approximate it with P-ET-R rather than using the mean of TWSC over the period when it was known.

**4) Figure A12: "The top line of represents precipitations datasets."  I did not get it.**

We apologize for this typo, a word was indeed missing making the sentence unintelligible.

**5) Page 24: "However, version 2.2 of this LSM, which assimilates GRACE data, performed poorly compared to its previous versions." There was an explanation: "Since this dataset assimilates GRACE measurements and was validated against GRDC observations, this may reflect overfitting of runoff that is better constrained than evapotranspiration, therefore leading to unrealistic ET values."  Could the author explain it more? Generally, the following seems more easy to understand: when runoff became more accurate, ET would be improved accordingly.**

We realized that our interpretation was maybe too superficial. Therefore, we deleted our explanation and instead suggested that this remains an open question for the community.

**6) Typo mistakes:**
**Pages 1, 21 and 23: The "Catchment Land Surface Model" was supposed to be CLSM as the abbreviation had appeared earlier.**
*It has been corrected.*

**"Fig. 2" and "Figure 2" were both used in the manuscript. Please unify the expression.**
Referring to the journal guidelines, it is advised to use Figure 2 when starting a sentence and Fig. 2 within a sentence.

---

## Author Response (AR1)

Fanny Lehmann
PhD student

fanny.lehmann@ens-paris-saclay.fr

Palaiseau, 12. October 2021

Hydrology and Earth System Sciences, The Editor

RE:     Water budget closure at the global scale

Dear Editor,

We are pleased to submit to *Hydrology and Earth System Sciences* a revised version of our manuscript titled "*How well are we able to close the water budget at the global scale?*".

We thank you and the reviewers for your comments and discussion that helped us improve the manuscript. Regarding the comments of the second reviewer, we clarified our choice of focusing on global runoff datasets in the data section 2.2.3 while mentioning at the end of the section 4.4 the results we obtained in 10 large basins while using GRDC runoff records. As explained in our detailed reply, the choice of Köppen-Geiger climate zones was based on the number of studies built on this classification and the lack of other variables providing a better explanation of the results.

Concerning the first reviewer's concerns about the processing of GRACE data, we discussed the misunderstanding of what they required us to modify. Several points raised related to a misunderstanding of the terminology used rather than a scientific issue. We have also detailed the known effects of time filtering in our detailed reply. We understand that the referees chose "major revision" for the paper. However, the changes required to the manuscript, based on the referees' comments, are not particularly extensive in terms of the scientific content. Following the advice from reviewer 2, we have removed several less important portions and removed any redundant information. As mentioned in correspondence with you by email, we could not identify issues raised by either referee that required substantial changes and all points are clarified and discussed in our detailed response to the comments.

If you feel that we missed something, please do not hesitate to get in touch.

We thank you in advance to consider our revised manuscript.

Sincerely,

On behalf of the co-authors

Fanny Lehmann

---

## Referee Report (RR1)

**Review of the first revision of hess-2021-279 - How well are we able to close the water budget on the global scale?**

By Fanny Lehmann et al.

Dear authors,

first of all, I would like to thank you for this comprehensive and very careful revision! Almost all of my comments and questions were answered satisfactorily and I appreciate the additional work and research that you have done for this revision. Furthermore, having published a similar (but less comprehensive and, in the meantime, outdated) study several years ago, I highly acknowledge the amount of work that you have put in the study. Evaluating global datasets and water budget closures in the absense of a true reliable reference is always a complicated task and I have to say, that you did a very good job in discussing and reasoning your results. It should be also appreciated that you have compiled a very helpful openly accessible repository, which is an important step towards a more open and reproducible science (and, by the way, should be mandatory for all publications in major journals).

**So, overall, I am happy to suggest a publication of the manuscript in HESS.** Furthermore, I would like to encourage the authors to continue their research on the water budget closure using global freely available datasets as this is highly relevant for many scientists.

**Minor comments:**

- If I were nitpicky, I would probably suggest to move some of the formulas (RMSD, time-filtering, etc.) to the appendix as they have been pulished in many other studies as well. This would further improve the reading flow of the manuscript.